# Northern Hemisphere ice sheets and ocean interactions during the last glacial period in a coupled ice sheet-climate model

Louise Abot[1], Aurélien Quiquet[2], and Claire Waelbroeck[1]

[1]LOCEAN/IPSL, Sorbonne Université-CNRS-IRD-MNHN, UMR7159, 75005 Paris, France
[2]LSCE/IPSL, CEA-CNRS-UVSQ, Université Paris-Saclay, UMR8212, 91190 Saint-Aubin, France

**Correspondence:** Louise Abot (louise.abot@locean.ipsl.fr)

**Abstract.** This study examines the interactions between the Northern Hemisphere ice sheets and the ocean during the last glacial period. Using the iLOVECLIM climate model of intermediate complexity coupled with the GRISLI ice sheet model, we explore the consequences of an amplification of the melt rates beneath ice shelves on ice sheet dynamics and the associated feedbacks. First, the amplification of oceanic basal melt rates leads to significant freshwater release from both increased calving
and basal melt fluxes. Grounding line retreat and dynamic thinning occur over the Eurasian and Iceland ice sheets, while the oceanic perturbation fails to trigger a grounding line migration over the coasts of Greenland and the eastern part of the Laurentide ice sheet. Second, similarly to hosing experiments with no coupling between the climate and the ice sheets, the influx of fresh water temporarily increases sea-ice extent, reduces convection in the Labrador Sea, weakens the Atlantic meridional overturning circulation, lowers surface temperatures in the Northern Hemisphere, especially over the North Atlantic Ocean,
and increases the subsurface temperatures in the Nordic Seas. Third, the freshwater release and latent heat effect on ocean temperatures lead to a decrease in ice sheet discharge (negative feedback) for the Greenland and Eurasian ice sheets. The Laurentide ice sheet does not feature significant volume variations in the experiments. On the one hand, the amplification of the shelf melt rates produces a weak perturbation due to low background temperatures and salinity at shelf drafts in the Baffin Bay and Labrador Sea in the model. On the other hand, the Laurentide ice sheet in the fully coupled model may be
overly stable. We show that we are able to force a grounding line retreat and a North American ice sheet volume decrease by imposing ad-hoc constant oceanic melt rates. However, in both sets of perturbation experiments, the Hudson Strait ice stream does not exhibit the past dynamical instability indicated by the presence of Laurentide origin ice rafted debris in the North Atlantic sediment records. This suggests that the model is too stable, specifically in the Hudson Bay region. Different ice sheet geometries or modelling choices regarding the basal dynamics beneath the ice sheet could help address this issue.
In summary, this study suggests that an episode of subsurface warming may trigger dynamical instabilities and ice discharges along the coasts of the Nordic Seas but subsequent ocean-ice sheet interactions may be characterized by negative feedbacks, thus dampening ice discharges. This study also emphasizes the need for further research using fully coupled models to explore the triggering mechanisms of massive iceberg discharges and to clarify the role of the ocean in these events.

# 1 Introduction

Recent studies of the Antarctic ice sheet show that warm oceanic water plays an important role in continental ice thinning, impacting not only floating but also grounded ice (e.g., Pritchard et al., 2012; Reese et al., 2018; Gudmundsson et al., 2019)). The increase of ocean sub-shelf melt rates induces a thinning and eventually break up of the ice shelves, reducing the buttressing effect. This results in an acceleration of the ice flow upstream and a dynamic thinning of the ice sheet inland. The ongoing warming of the ocean (Johnson and Lyman, 2020) could lead to the collapse of the West Antarctic Ice Sheet, among other dramatic consequences, amplifying the sea level rise (Joughin and Alley, 2011; Naughten et al., 2023). To the north, the Greenland ice sheet's last floating ice tongue also undergoes a thinning driven by warmer ocean temperatures (Wekerle et al., 2024). Similarly, a number of studies shows that ocean subsurface warming during the last glacial period could have destabilized ice sheets and led to massive iceberg discharges (e.g., Shaffer et al., 2004; Marcott et al., 2011; Max et al., 2022). Indeed, it was suggested that massive iceberg discharges occur after an initial reduction of the Atlantic Meridional Overturning Circulation (AMOC) and the build up of a heat reservoir at the subsurface in the North Atlantic from both observations and modelling studies (e.g., Jonkers et al., 2010; Max et al., 2022; Mignot et al., 2007).

The Last Glacial Period refers to the portion of the last glacial cycle comprised between the end of the last interglacial period and the transition into the present interglacial. It extends from about 75 kyr before present (BP) (e.g., Martinson et al., 1987) to about 15 or 12 kyr BP (e.g., Rasmussen et al., 2014). An intriguing feature of the last glacial period is its millennial-scale variability, first revealed in Greenland ice cores, known as the Dansgaard-Oeschger (DO) cycles (Dansgaard et al., 1984, 1993). These are transitions between relatively cold (stadial) and warm (interstadial) periods in the Northern Hemisphere (Johnsen et al., 1992). The rapid warming (or DO event) generally takes place over a period of several decades and is of the order of $10^oC$ at the North Greenland Ice Core Project (NGRIP) site (Kindler et al., 2014). Such events occurred repeatedly during the last glacial period. They are followed by a slower cooling of the order of a hundred to a thousand years, and the cycle ends with a sudden cooling (Kindler et al., 2014).

Some of the stadials are accompanied by massive iceberg discharges from the Northern Hemisphere ice sheets in the North Atlantic Ocean, at a frequency of several thousand years (Heinrich, 1988; Bond and Lotti, 1995). The iceberg discharges are identified in marine sediment cores by one or several layers of ice rafted debris, the materials entrapped and transported by drifting ice (Bond et al., 1992; Bond and Lotti, 1995; Elliot et al., 1998). The iceberg discharges are referred to as Heinrich Events (HE) when the continental ice is originating from the Laurentide ice sheet, the central and eastern parts of the North American ice sheet. The Heinrich events mostly occur after several DO cycles and are followed by a DO event (Bond et al., 1993). Barker et al. (2015), through the examination of a site southwest of Iceland, suggested that massive iceberg discharges were a consequence of prolonged stadial conditions.

Nonetheless, there is no scientific consensus on the ultimate cause behind the massive iceberg discharges to date. For instance, some studies have shown that self-sustained ice sheet oscillations could be at play (MacAyeal, 1993), while others propose that the discharges were triggered by an external forcing such as a change in ocean conditions (Alvarez-Solas et al.,

2010). The focus of our study is to test the latter hypothesis, using a coupled climate-ice sheet model, namely iLOVECLIM-GRISLI.

More specifically, several studies show that a rise in subsurface ocean temperature during a stadial in the North Atlantic could have led to massive iceberg discharges (e.g., Shaffer et al., 2004; Marcott et al., 2011; Max et al., 2022). Alvarez-Solas et al. (2010) have shown that fluctuations of the oceanic temperatures could have triggered Northern Hemisphere ice sheets surges (i.e., an unsteady state of the ice stream associated with rapid ice flow and dynamic thinning of the ice sheet inland) and iceberg discharges during the last glacial period. Forcing the Laurentide ice sheet and Hudson stream with varying basal melt rates results in the collapse of buttressed ice shelves and subsequent acceleration of inland ice streams (Alvarez-Solas et al., 2013). Isostatic adjustment may then allow the regrowth of the ice sheet (Bassis et al., 2017). Modelling results show that ocean temperature increases can also trigger or amplify ice discharges from the Eurasian and Greenland ice sheets (Alvarez-Solas et al., 2019; Tabone et al., 2019). However, in these studies, the ocean is considered as a forcing and as such does not allow to fully capture the interplay between ocean and ice sheets. The novelty of this study lies in the use of a climate-ice sheet model : the iLOVECLIM model of intermediate complexity coupled to the GRISLI ice sheet model.

Indeed, grasping the response of the entire ice sheet to an oceanic perturbation, or the subsequent oceanic changes following an influx of continental water, is complex due to the presence of numerous feedback mechanisms at the ocean-ice sheet interface (Goosse et al., 2018; Holland et al., 2020). For instance, under warming conditions, initial ice sheet melting releases cold and fresh water at the surface, which can lead to negative feedback on climate (Swingedouw et al., 2008; Li et al., 2024). Additionally, ocean-driven ice shelf thinning may raise the depth-dependent freezing temperature at the ice–ocean interface, reducing oceanic basal melt rates, that is another negative feedback (Van Achter et al., 2023). Following an oceanic warming, the shelves melt water release into the ocean could also lead to decrease vertical mixing and favor sea ice expansion, reducing the initial ocean warming and basal melt rates, thus inducing a negative feedback as well (Goosse et al., 2018). There are also numerous possible positive feedbacks that could contribute to the destabilization of the grounded ice sheet. For instance, freshwater release can increase surface stratification, leading to transient subsurface warming that amplifies oceanic basal melting (Moorman et al., 2020; Li et al., 2024). Another positive feedback is the marine ice sheet instability, which is linked to the dynamics of the grounding line. The mathematical formulation of the ice flux at the grounding line shows that this flux increases with the ice thickness (Schoof, 2007; Tsai et al., 2015). When a marine-terminating ice sheet sits on an upward-sloping bed toward the ocean, an initial retreat of the grounding line causes ice thickening and acceleration of the ice flow at the grounding line, which in turn drives further retreat. This process may have played a role in Heinrich events given the bedrock's shape at the Hudson Strait mouth (Schoof, 2007). Additional feedback mechanisms could emerge from ice sheet-cavity scale processes, such as tidal forcing or buoyancy-driven circulation beneath ice shelves (e.g., Makinson et al., 2011; Gwyther et al., 2016). For example, a rise in ocean temperature at the base of the ice shelves could enhance the meltwater production, thereby strengthening the circulation within a cavity, leading to an increased heat supply and further melting (Gwyther et al., 2016). However, uncertainties related to these processes remain large since very few cavity-enabled ocean models can operate at the global scale. For this reason and given our coarse resolution ocean model, we do not discuss the cavity-scale feedbacks any further in this paper.

In our setup, under constant or transient forcings corresponding to the last glacial period, the iLOVECLIM-GRISLI coupled model does not exhibit abrupt internal oscillations, neither in the ice sheets nor in the ocean circulation. Therefore, in this study, we chose to conduct perturbation experiments at the ice-ocean interface. With the fully-coupled model, we can examine whether a perturbation at the interface is being amplified or dampened.

Here, we take advantage of the fully-coupled model to address the following questions : How does ocean subsurface warming at characteristic ice shelf drafts affect the ice sheets dynamics? How does the continental ice melt release affect in turn oceanic circulation, hydrography and ice sheet dynamics? In this work, we impose amplified oceanic melt rates to the 40 kyr BP Northern Hemisphere ice sheets to get insights on both past climate variability and mechanisms at play at the ocean-ice sheet interface.

## 2   Model and methods

### 2.1   Climate and ice sheet model

#### 2.1.1   iLOVECLIM

For this study, we use the coupled climate model of intermediate complexity iLOVECLIM which is a code fork of LOVECLIM version 1.2 (Goosse et al., 2010). The core of the model includes oceanic, atmospheric and vegetation components (namely CLIO, EcBilt and VECODE). The model has been used for a wide range of climate studies, from the last million years to the Holocene (e.g., Caley et al., 2014; Bouttes et al., 2018; Arthur et al., 2023). We remind here some of its main features. VECODE is a dynamic vegetation and carbon allocation model. CLIO is a free surface ocean general circulation model solved on a spherical grid with $3^o$ latitude and longitude resolution and 20 irregular levels in z-coordinates. It includes a thermodynamic-dynamic sea-ice component (Fichefet and Maqueda, 1997). Despite its coarse resolution, CLIO is able to represent large-scale features. Taking the example of the North Atlantic Ocean, these features include the main gyres (North Atlantic and subpolar gyres), as well as the main currents. The Gulf Stream, the North Atlantic Current, the Irminger Current and the East Greenland Current are depicted, although the latter are wider than in higher resolution models (Supplementary S1). EcBilt is a quasi-geostrophic atmospheric model solved on a T21 spectral grid with a resolution of $5.6^o$ in both latitude and longitude. It includes 3 vertical levels at 800, 500 and 200 hPa. Given that precipitation biases can alter the performance of CLIO to reproduce large-scale oceanic circulation, our model includes a flux correction of the precipitation rates since Goosse et al. (2010). Specifically, precipitation is enhanced over the North Pacific while reduced over the Arctic and Atlantic ocean (Supplementary S2). Major model parameters are listed in Supplementary Tables S3a-b. We use the default values from the iLOVECLIM source archive, except for the Bering Strait scaling factor and the greenhouse gas (GHG) concentration radiative forcing factor. The first is set to zero to ensure a closed Bering Strait. The second is doubled (Timm and Timmermann, 2007) to artificially increase the climate sensitivity and to achieve a relatively cool climate and maintain the ice sheets at 40 kyr BP. This modelling choice is informed by the relatively low equilibrium climate sensitivity of iLOVECLIM, which is approximately $2 \ ^oC$(Bouttes et al., 2024), in comparison with the IPCC AR6 best estimate of $3 \ ^oC$ and likely range of $2.5$ to $4 \ ^oC$ (IPCC, 2023).

### 2.1.2 GRISLI

iLOVECLIM is coupled to the dynamic ice sheet model GRISLI, which is a 3D thermomechanically coupled ice sheet model that is used here on a cartesian grid of the Northern Hemisphere at 40 km resolution. GRISLI is an appropriate tool to address ice sheet dynamics and interactions with the oceanic component as it explicitly calculates ice stream velocities, the position of the grounding line and the behavior of ice shelves. For instance, GRISLI has shown a good ability to reproduce grounding line migrations and ice volume changes in Antarctica during the last 400 kyr BP (Quiquet et al., 2018a; Crotti et al., 2022). The equations are described in Ritz et al. (2001) and Quiquet et al. (2018a). They are written under shallow ice and shallow shelf approximations. Major model parameters are listed in Supplementary S3c.

The deformation is calculated using the Glen flow law with an enhancement factor ($E_f$) to account for anisotropy.

When the ice sheet's base is at the pressure melting point, the basal drag is a linear function of the basal velocities over the grounded ice sheet (Weertman, 1957). When the ice sheet's base is below the pressure melting point, the sliding velocity is set to zero (infinite friction). The friction is set to zero for the floating ice shelves. Following the Weertman law, the basal drag is equal to the opposite of the basal sliding velocity times a basal drag coefficient $\beta$. This coefficient $\beta$ is set to the effective pressure at the base of the ice sheet times a tuning parameter ($C_f$) and is bounded by a minimum ($\beta_{min}$) and a maximum ($\beta_{max}$) value. Over thick sediment layers (more than 200 m) the basal drag is reduced by a factor 20. We use the soft-hard basal drag mask (i.e. the sediment map) from Laske (1997) and the geothermal heat flux from Shapiro and Ritzwoller (2004).

The calving of icebergs is not modeled explicitly. Instead, the floating ice calves when its thickness falls below a critical thickness threshold, which varies with oceanic depth (Supplementary S4). This way, the shelves can not extend over the deep ocean. It is to be noted, that after calving, the ice shelves are allowed to grow back, as a result of upstream ice flow and surface accumulation.

The grounding line formulation follows Tsai et al. (2015), and its position is determined by interpolating between the last grounded and the first floating grid cells.

Glacial isostatic adjustment is accounted for using an elastic-lithosphere-relaxed-asthenosphere model with a relaxation time of 3000 yr (Le Meur and Huybrechts, 1996).

### 2.1.3 Coupling

The coupling procedure is described in Roche et al. (2014) and Quiquet et al. (2021b). In iLOVECLIM, the atmospheric model ECBilt includes an online downscaling (Quiquet et al., 2018b). At each atmospheric time step, temperature and precipitation are computed on the finer resolution GRISLI grid using the standard ECBilt energy and moisture equations. This subgrid information is fed into an Insolation Temperature Melt model (Pollard, 1980; Van Den Berg et al., 2008) to compute surface mass balance every four hours. This time step allows for numerical model stability, as the ECBilt model does not resolve the diurnal cycle. The yearly average surface mass balance and surface temperature are then used by GRISLI. In turn, ECbilt receives the yearly topography and ice mask data from GRISLI.

The continental ice melt fluxes are computed in GRISLI each year. These are distributed to the oceanic model over the following year. For the grounded ice sheet, surface melt is routed to the nearest oceanic grid cell following surface topography while the oceanic basal melt and the calving flux are transferred at the ocean surface where they occur. The local latent heat flux associated with the calving flux and that corresponds to the melting of icebergs by the ocean, is taken into account and the ocean temperatures are adjusted. More details about the freshwater coupling are given in Supplementary S2.

The oceanic basal melt rate (OBM) is parameterized as a quadratic function of ocean temperatures (Holland et al., 2008; Favier et al., 2019). When the local ocean temperature is above the freezing temperature, the oceanic basal melt rate is derived as:

$$OBM = \gamma_T * A * (T - T_f)^2 \tag{1}$$

where the local freezing temperature $T_f = T_f(S, z = 200\ m)$ is a function of salinity, following the formulation of Millero (1978). The freezing temperature is here computed at a characteristic shelf draft of 200 m. $A = (\frac{\rho_0 C_{p0}}{L\rho_i})^2\ K^{-2}$ is a constant depending on physical parameters, with $\rho_0 = 1030$ kg/m$^3$ and $C_{p0} = 4002$ J/kg/K the density and specific heat capacity of seawater, $L\rho_i = 300.33\ 10^6$ J/$m^3$ the latent heat of fusion of ice times the ice density. $\gamma_T$ is a tuned parameter that accounts for the heat transfer velocity, and that has been calibrated for this study to $1.09 * 10^{-5}$ m/s. This value for $\gamma_T$ ensures that the ice sheet volume remains intermediate between that of the Pre-Industrial (PI) period and the LGM, preventing excessive expansion or disintegration. The OBM is computed in the ocean model each day and integrated over each year as the time step for the dynamic ice sheet model is annual. For each horizontal grid point, the rate is calculated for all vertical levels in the ocean model, and the ice sheet model retains only the value corresponding to its actual shelf draft. When the ice sheet model cell is not covered by the ocean model, the value of the nearest ocean cell is retained.

The fluctuations in ice sheet volume have no impact on the global volume of the ocean in the model. As a result, sea level and bathymetry in the ocean model remain unchanged.

## 2.2 Experimental setup

### 2.2.1 Initial condition

We conduct a long coupled ice sheet-climate simulation under constant external forcing to derive the initial state. This approach assumes climate equilibrium, which is not realistic. However, the relatively small variations in greenhouse gas concentrations and insolation at 65 $^o$N around 40 kyr BP support the use of an equilibrium simulation for this specific time slice. Furthermore, this period aligns with Heinrich Stadial 4 (HS4), which extends from 39.85 to 38.17 kyr BP (Waelbroeck et al., 2019).

The climate model is forced with insolation (Berger, 1978) and greenhouse gases concentration (Lüthi et al., 2008). Forcings are held constant at their 40 kyr BP values (namely 0.013$^o$, 23.61$^o$ and 0.004 for eccentricity, obliquity and precession index, 220 ppm for the $CO_2$ concentration). The ice sheet model is forced with sea level reconstruction (-64 m at 40 kyr BP; Waelbroeck et al., 2002). The latter is only used by GRISLI to determine which areas of land are located above the sea level (i.e., where the continental ice can grow). The long spin-up starts from a previous equilibrium simulation of the Last Glacial

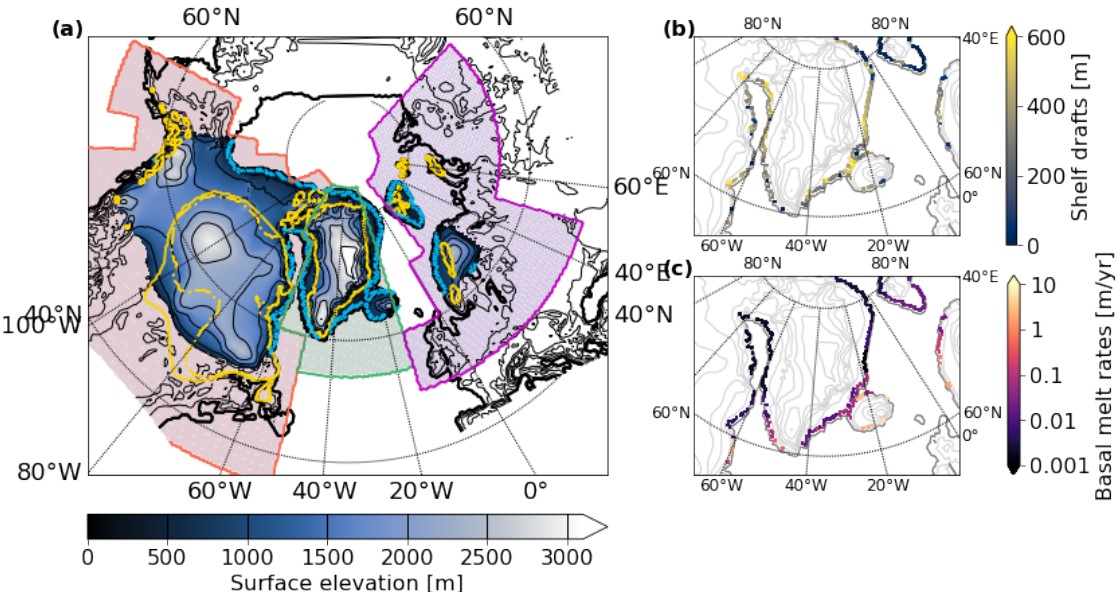

**Figure 1.** 40 kyr equilibrium. Ice sheet elevation (shades of blue) and elevation above sea level (shades of gray) [m] at the end of the equilibrium simulation. Light blue contour is the mean position of the grounding line at the same time. Yellow contours are the minimal and maximal ice sheets extent in the reconstruction from Gowan et al. (2021), these differ only over the central North American ice sheet. Areas used to derive regional ice volumes on Figure 2 for North American, Greenland-Iceland and Eurasian ice sheets are contoured in pink. (b) Ice shelf drafts [m] and (c) basal melt rates beneath the shelves [m/yr] at the end of the equilibrium simulation.

Maximum (LGM) similar to the one described in Quiquet et al. (2021b). We use a LGM bathymetry for the ocean model (Lhardy et al., 2021). However, since sea level and bathymetry at 40 kyr BP have intermediate value between the LGM and the Pre-Industrial values, we also test the sensitivity of the results to the Pre-Industrial bathymetry in the discussion section. Thus, two extreme bathymetry possibilities are investigated.

For this long spin-up simulation, we use an asynchronous coupling between the ice sheets and the rest of the climate system. Since the ice sheets feature slower dynamics than the ocean and the atmosphere, we run the ice sheet model for 10 years every year of the rest of the climate model. The spin-up simulation runs for 8,000 years for the ocean and the atmosphere but 80,000 years for the ice sheets. During the spin-up phase the freshwater fluxes from the ice sheet model to the ocean are ignored. This approach is necessary because we cannot conserve both total ice mass change and the rate of mass change when using an acceleration factor (Supplementary S2). However, at the end of the spin-up phase the ice sheet is in equilibrium with the rest of the climate system. Therefore the ice sheet volume is quasi stable and produces a net zero freshwater flux to the ocean. There is thus no significant discontinuity in our control experiment with ice sheet freshwater flux included when branched from the spin-up simulation.

At the end of the spin-up simulation, we obtain a total volume of 24 million km$^3$ for the Northern Hemisphere ice sheets, which corresponds to 48 mSLE (mean sea level equivalent; directly calculated from the volume of grounded ice that is above

sea level and compared to the present-day ocean area). In comparison, the reconstruction from Gowan et al. (2021) presents volumes of around 12 and 16 million km$^3$ for minimal and maximal scenarios, corresponding to 28 and 37 mSLE respectively.

However, Gowan et al. (2021) acknowledge that their total ice sheet reconstructions for the period between 57 and 29 kyr BP (MIS3) do not align with proxy-based sea level reconstructions. Specifically, they estimate a global sea level that is 20 m higher (maximum scenario) than the reconstructed value of approximately $-60$ to $-90$ m at 40 kyr BP (Waelbroeck et al., 2002; Arz et al., 2007; Siddall et al., 2008). Also, the evolution of the Northern Hemisphere ice sheets remains uncertain prior to the Last Glacial Maximum. Reconstructions often rely on inverse methods that use observational data and estimates of global mean sea

level, sometimes supported by numerical ice sheet modelling (e.g.; Marshall et al., 2000; Kleman et al., 2010; Pico et al., 2018; Gowan et al., 2021; Dalton et al., 2022). Significant discrepancies exist between the different reconstructions. For example, the question of whether the Hudson Bay area was glaciated around 40 kyr BP and to what extent remains debated (e.g.; Batchelor et al., 2019; Miller and Andrews, 2019). Given these uncertainties, we conclude that our simulated ice sheet volume falls within an acceptable range for the 40 kyr BP time slice.

The ice sheet conditions at the end of the 40 kyr BP equilibrium simulation are depicted in Figure 1. Ice is located over North America, Greenland, Iceland, Svalbard and Fennoscandia. Our Northern American ice sheet is larger than the one presented in Gowan et al. (2021) for the same period. In contrast to this study, we obtain a glaciated Iceland and a marine terminating Fennoscandian ice sheet. Also, the DATED-1 reconstruction for the Eurasian ice sheet (Hughes et al., 2015) presents a reduced extent of the Fennoscandian ice sheet and no Svalbard ice sheet compared to our simulated ice sheet.

However, this reconstruction corresponds to the 38–34 kyr BP period, which is slightly later than the time slice examined in this study. The ice shelf area of our simulated ice sheet is relatively small, with no extensive regions that are entirely covered by ice, in contrast to what is observed in Antarctica today.

### 2.2.2 Perturbation experiments

Each of the following experiment is branched on the 40 kyr BP equilibrium. External forcings are held constant at their 40

225 kyr BP values. Differing from the spin-up simulation, the climate and the ice sheets are annually coupled. The acceleration is only used for the spin-up simulation, in order to equilibrate the ice sheets with the climate, as the ice sheets take a long time to adjust. Here, freshwater fluxes from the ice sheet to the ocean are taken into account. The control experiment (CTRL) is the continuation of the equilibrium simulation described in 2.2.1 but with addition of freshwater flux to the ocean resulting from ice sheet melting.

Each perturbation experiment consists of multiplying the oceanic basal melt rates (defined in Eq. 1) at each grid point by an amplification factor X, with X ranging from 5 to 300, for 500 years. This scaling (X) of the basal melt is equivalent to imposing a temperature anomaly, that depends on the water masses and background state of the ocean, at the shelf drafts. Specifically, we apply a perturbation whose intensity is proportional to the local temperature anomaly with respect to the freezing temperature, $T_f$. Indeed, when $T > T_f$, adding a perturbation term $\delta T$ to T defined by $\delta T = \epsilon(T - T_f)$ (with $\epsilon > 0$) leads to the following

oceanic basal melt rate : $OBM = \gamma_T A(T + \epsilon(T - T_f) - T_f)^2 = \gamma_T A(1+\epsilon)^2(T - T_f)^2$. And we rewrite X$= (1+\epsilon)^2$. So there is a direct correspondence between X and the temperature anomaly intensity $\epsilon$ seen by the ice shelves. For example, a doubling

of the temperature anomaly with respect to the freezing point corresponds to $\epsilon = 1$ and $X = 4$. A 10 time increase of the temperature anomaly corresponds to $\epsilon = 9$ and $X = 100$.

This way, the perturbation experiments aim to represent an increased heat flux to the ice shelf drafts, reflecting subsurface ocean warming in the North Atlantic during a Dansgaard-Oeschger (DO) stadial (e.g., Rasmussen and Thomsen, 2004). The perturbation experiments were designed to be spatially consistent with the physics of the water masses. For instance, our design allows us to account for the fact that an abrupt change in AMOC likely leads to temperature changes in the AMOC's main areas of influence, rather than a uniform temperature change over the North Atlantic and Nordic Seas.

Here, we amplify the basal melt rates instantaneously, as this seems to be the most direct way to maximize the ice sheet response to the perturbation and to evaluate the sign of the feedbacks at the ice-ocean interface. Note that the impact of a gradual increase in basal melt rates on ice loss is similar to that of an instantaneous increase (Supplementary S5). Still, a limitation of the basal melt scaling approach is that the experiment results are dependent on the initial representation of the temperature in the ocean and model biases. To overcome this limitation, we perform additional experiments by applying a constant temperature offset within the formulation of the basal melt rate (section 4.1).

From the ice sheets point of view, increasing the oceanic basal melt rates is equivalent to imposing a subsurface warming yet this has no effect on initial ocean temperatures. Initially, the ocean structure is the same for the control and perturbation experiments, as the simulations are branched on the same spin-up. Since we use a coupled setup, the ice sheet retreat induced by the perturbation impacts the ocean through the resulting freshwater flux. The ocean model adjusts its temperatures and salinity in response to the freshwater input, with associated impacts on the density profile, vertical convection and hence ocean circulation. Thus, the ocean model response can modulate the perturbation in the vicinity of the ice shelves. Therefore in our perturbation experiments the freshwater feedback that could arise from the ocean-ice sheet interactions is taken into account, unlike traditional hosing experiments.

## 3 Results

### 3.1 Ice sheet response

The oceanic perturbation causes a decrease in the total volume of the Northern Hemisphere ice sheets. The relative contributions of the regional ice sheets show that the North American one contributes the least to the total volume change, while the Greenland, Iceland and Eurasian ice sheets contribute the most (Figure 2). The volume of the Greenland-Iceland ice sheets decreases rapidly over the first hundred years before reaching a new state with minimal ice volume change for experiments with perturbation factors above 20. The Eurasian ice sheet loses volume throughout the perturbation period for all experiments. Once the perturbation is halted, Eurasia and Greenland regain mass at a slower pace. The North American ice volume remains relatively unaffected by the oceanic disturbance. For the highest perturbation factors, it even increases slightly till $\sim 500$ years and decrease once the perturbation is halted (Figure 2a).

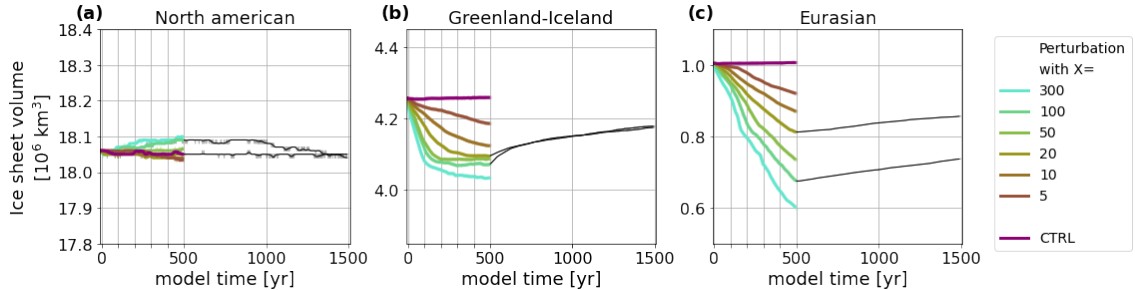

**Figure 2.** Continental ice volume response to oceanic perturbation (i.e. amplified OBM by a coefficient X). Time series of grounded ice sheet volume [$10^6$ km$^3$] for the (a) North American, (b) Greenland and Iceland and (c) Eurasian ice sheets in experiments with different perturbation factors. The thin black lines are the free evolution, once the perturbation is released, of the experiments where X = 20 and 100. Note that the same y-axis spacing is applied on the plots (0.6 $10^6$ km$^3$).

Despite its low resolution (40 km), GRISLI is capable of representing large-scale features such as major ice streams at the
LGM (e.g., Quiquet et al., 2021a). The major ice streams, such as the Hudson Strait, Lancaster Sound, and Amundsen Gulf ice streams (nomenclature follows Margold et al., 2015), remain present at 40 kyr BP (Figure 3a). Near the southern margin of the North American ice sheet, on the contrary, there are no well-identified ice streams. There are also large velocities (above 500 m/yr) along the south east coast of Greenland and around Iceland (Figures 3a,b). Active regions with weaker yet still above 100 m/yr velocities also include both Greenland coasts and the Fennoscandian coast.

The ice sheet evolution in the experiment with a large perturbation factor is depicted in Figures 3c-f. After 200 years, the thinning is significant on the coasts bordering the Nordic Seas and to the south of the Greenland ice cap (Figure 3c), corroborating the volume time series (Figure 2). A smaller thinning is located inland to the south of Greenland. Minor thickening ($< -100$ m ; note that the colorbar scale for Figure 3c,e,f is logarithmic) occurs over the Greenland ice sheet, especially along the east coast, upstream of coastal thinning. The latter is associated with the largest velocity decreases after 200 years ($< -100$
m/yr ; yellow contour on Figure 3d). In this area, the perturbation fails to trigger a grounding line retreat. In contrast, ice flow velocities increased with the grounding line inland retreat on the Fennoscandian, Svalbard and Iceland coasts, sometimes until there is no more connection between ocean and land as it is the case for the Iceland after 200 years (blue contours ; Figures 3b versus d). These contrasting responses reflect regional differences in the initial basal met rates (Figure 1c) that arise from regional differences in ocean temperature and salinity (through freezing temperature) at the shelf drafts, as we shall see in
section 3.2.

To get further insight on the nature of the volume variations, we decompose the ice thickness changes such as (Eq. 2; mass conservation) :

$$H(t) - H(0) = \int_0^t (SMB - BM - C)dt - \int_0^t \boldsymbol{\nabla} \cdot (\boldsymbol{u}H)dt \qquad (2)$$

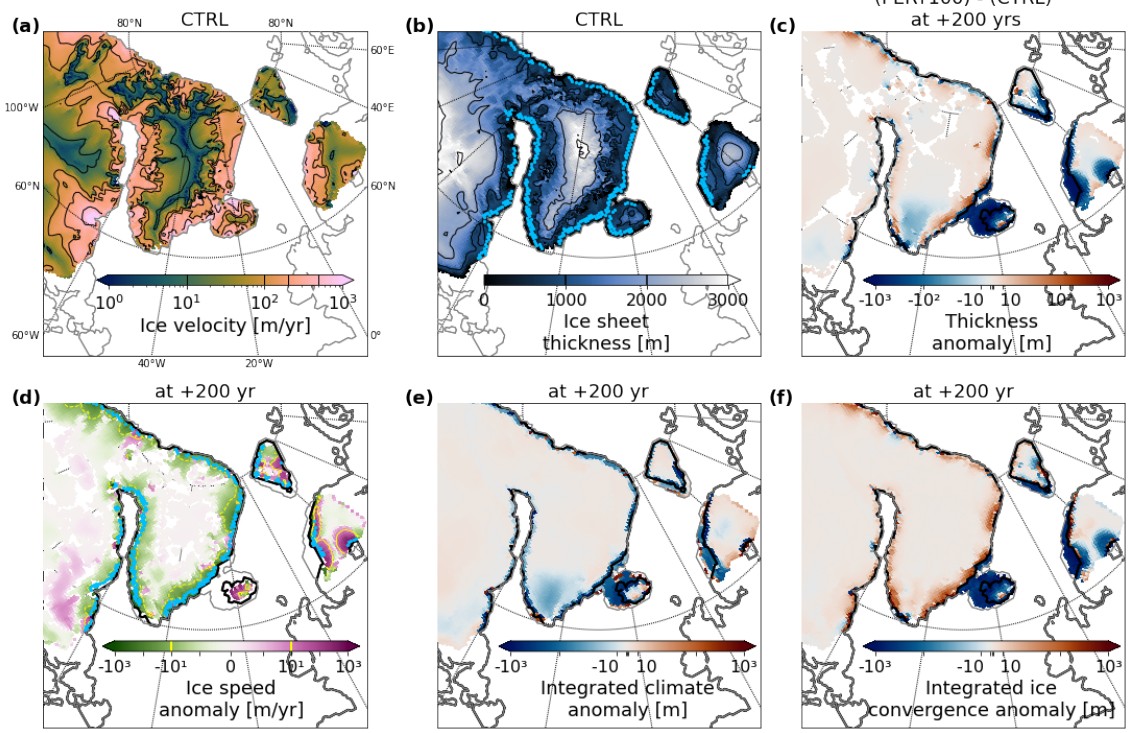

**Figure 3.** Ice sheet history in perturbation experiment with factor $X = 100$. Ice sheet (a) velocity [m/yr] and (b) thickness [m] averaged over the control simulation. Sky blue contour indicates the position of the grounding line. PERT100 - CTRL anomalies of (c) thickness, (d) ice velocities, (e) thickness due to climate anomalies, (f) due to ice convergence, after 200 years. Only significant values to one standard deviation of the control are shown for (c) and (d). Please note that the colorbar for the ice sheet thickness is linear while all the other ones are logarithmic.

with $H$ the ice thickness, *SMB* the surface mass balance, *BM* the basal melt rate, *C* the calving rate and $\nabla \cdot (\boldsymbol{u}H)$ the ice flow divergence, *dt* is one year. We refer to the first term as climate term (although it contains the calving which is not strictly speaking a climate term). The residual is the lowering or thickening due to ice flow (dynamical effect).

The perturbation induces a loss of volume both through climate and dynamic thinning (Figure 3e,f). The oceanic melt rate perturbation is visible through negative thickness anomalies along the coasts on Figure 3e. The inland thinning over Greenland is also caused by the climate term. The latter is due to a change in surface mass balance (less accumulation over the Greenland when the fresh water is released; Figure S6).

The thinning due to the dynamic term is most important in areas such as the west coast and southern part of the Fennoscandian (proglacial lake grounding line instability for the latter), the Svalbard, Iceland and southern tip of the Greenland ice sheet. Also, the minor thickening in coastal areas is due to the dynamic term. In the model, the ice flux at the grounding line connects the ice thickness and velocities in the grounding zone through a power-law relationship (Tsai et al., 2015). Where the

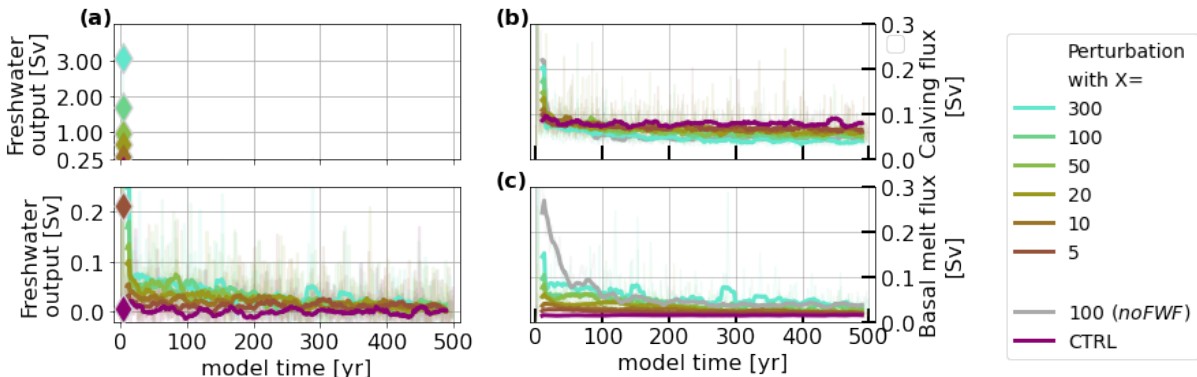

**Figure 4.** Time series of (a) freshwater output from GRISLI to iLOVECLIM [Sv] (as a total ice volume change) in experiments with different perturbation factors. Diamonds are the freshwater release for the first year of perturbation, light colors are the yearly outputs and saturated colors are smoothed time series (running mean over 21 years, centered). Times series of (b) calving flux [Sv] and (c) basal melt flux [Sv], also smoothed (running mean over 21 years, centered).

oceanic perturbation fails to trigger a grounding line retreat, the perturbation only causes a reduction in thickness that leads to a reduction of the ice flux and to upstream convergence (Figure 3f). However, the resulting ice elevation increase in these areas is insufficient to produce a discharge through an increase of the ice sheet surface slope. This thickening effect also appears north of the North American ice sheet, at its connection with the Arctic ocean, and east at its connection with the Baffin Bay. Together with positive anomalies of the surface mass balance to the south, this contributes to the small thickening of the North

American ice sheet for experiments with at least a 10 time increase of the temperature anomaly with respect to the freezing temperature, at the shelf drafts ($X \geq 100$ ; Figure 2).

     Still, the overall volume loss experienced by the ice sheets is equivalent to a freshwater release toward the ocean (Figure 4a). This freshwater flux is particularly large during the first few years in all experiments and results from both abrupt increased calving and increased basal melt fluxes in the first few years (Figure 4b,c). The initial amplification of the melt rates leads

to large losses at the beginning as a basal melt flux (Figure 4b). The initial amplification of the melt rates also leads to an abrupt increase of the calving flux (Figure 4c). Indeed, some of the shelves are rather thin before the perturbation is triggered (Figure 1b), so the perturbation causes these shelves' thickness to fall below the calving threshold (Supplementary S4). Then, the calving flux drops toward the control value after 10 years while the basal melt flux slowly decreases toward the control value. This results in a freshwater flux that decreases with time (but remains significant). This decrease is partly attributed to

the reduced ice volume exposed to basal melting as ice shelves thin or vanish in certain regions, such as Svalbard and Iceland. It also suggests the presence of ice sheet stabilizing mechanisms within the model, that is, a negative feedback that mitigates ice volume loss. Here, this feedback involves a decrease of the temperature at shelf drafts, which in turn reduces the basal melt rates.

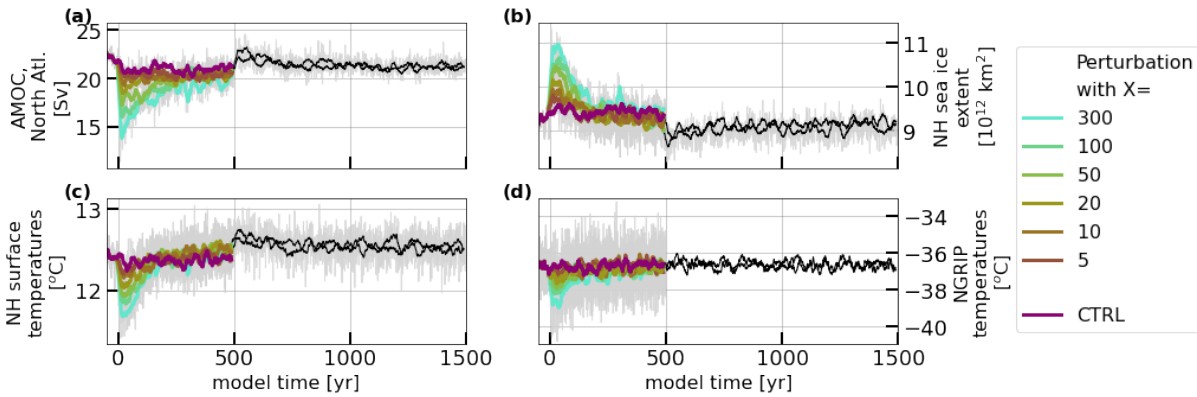

**Figure 5.** Times series of (a) AMOC intensity in the North Atlantic [Sv], (b) Northern Hemisphere sea ice extent [$10^{12}$ km$^2$], (c) Northern Hemisphere and (d) NGRIP surface temperatures [$^o$C] in perturbation experiments with different factors. The thin black lines represent the unperturbed extension of the simulations with factors 20 and 100. Light colors are the yearly outputs and saturated colors are smoothed time series (running mean over 21 years, centered).

At this point, this study suggests that an episode of subsurface warming may have triggered dynamical instabilities for the Eurasian, Svalbard and Iceland ice sheets. However, after about a decade, the simulated freshwater release to the ocean decreases with time. The freshwater release has the ability to modify climate, ocean hydrography and circulation which can in turn affect the ice sheets. In the following section, we investigate climate and ocean changes and examine how they are connected with regional ice sheet changes.

## 3.2 Climate and ocean changes

Subsequent to the continental melt water release, the Atlantic meridional overturning circulation (AMOC) weakens for several years in all the simulations. There are no off modes in our simulations even with the largest perturbation factors. At the same time, the extent of the sea ice cover increases and the surface temperatures decrease over the Northern Hemisphere (Figure 5). The AMOC rapid decrease is followed by a slower recovery toward the control simulation values, corroborating the reduction of the freshwater inflow with time (Figure 4a).

Although one could expect that the transient Northern Hemisphere cooling would allow ice sheet regrowth during the perturbation period, this is not the case. Indeed, the Northern Hemisphere surface cooling is essentially located over the North Atlantic Ocean (especially over the Nordic and Labrador Seas), while only limited cooling occurs over Greenland and Europe, and no notable temperature changes take place over North America.

For most of these metrics, there are no past estimates available. Greenland temperatures at the NGRIP sites is the only exception (Kindler et al., 2014). At this location, the authors obtain around -40 to -45 °C around 40 kyr BP, while we simulate warmer temperatures around -35 to -40 °C. Paleoproxy records show that the AMOC was in a much weaker regime during

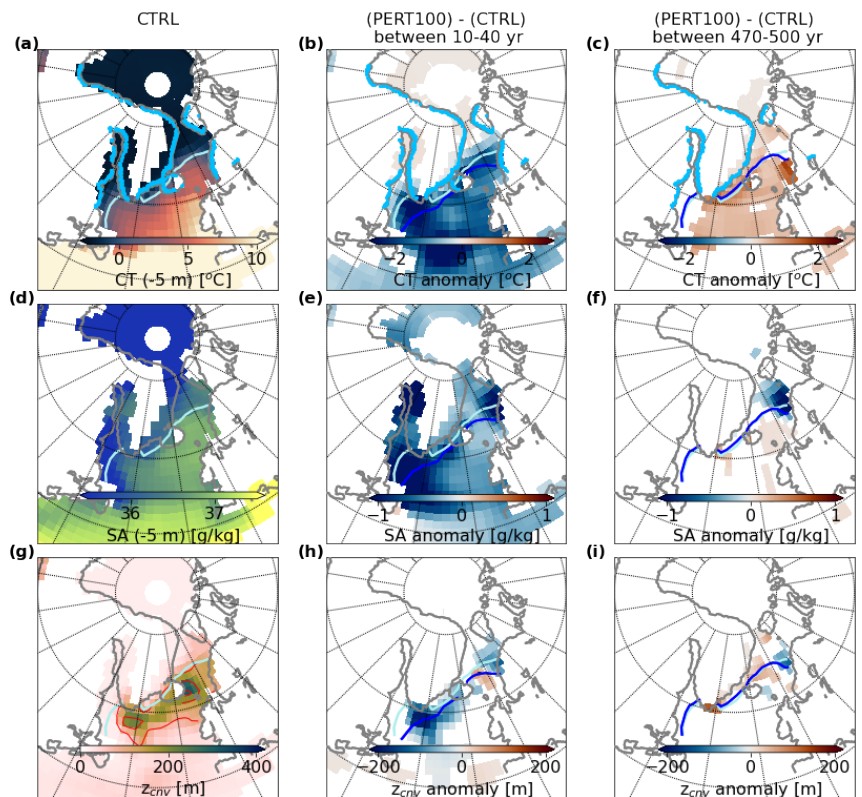

**Figure 6.** Hydrography at 5 meters depth horizon, in perturbation experiment with factor $X = 100$. Maps of (a)-(c) conservative temperature [$^o$C] over the control simulation and associated anomaly over 10-40 years and over 470-500 years comparing PERT100 and the control simulation. (d)-(f) Same for practical salinity [psu]. (g)-(i) Same for convection depth [m]. Only significant values to one standard deviation of the control simulation are shown for anomalies. Sky blue contour is the grounding line position in the control, after 40 years and after 500 years (resp. on a, b, c). Turquoise and dark blue contours are annual 30% sea ice contours in the control and the perturbation experiment, respectively.

stadials than interstadials but do not provide quantitative estimates in Sv (Gottschalk et al., 2015; Henry et al., 2016; Waelbroeck et al., 2018).

Differences in timings of the different variables are difficult to establish as the times series are noisy. Considering smoothed times series (21 years centered running mean), the maximum changes for the sea-ice extent ($+10^{12}$ km$^2$ in comparison with the control), the AMOC ($-5$ Sv), the NGRIP ($-0.53$ $^oC$) and Northern Hemisphere ($-0.49$ $^oC$) temperatures are reached after 14, 18, 21 and 33 years respectively for the experiment with a large temperature anomaly intensity at the shelf drafts (i.e. X=100 ; Figure 5).

In the unperturbed simulation, convective areas are located south of the sea-ice edge in the Labrador Sea and south of the Nordic Seas (Figure 6g). Following the freshwater release, major cooling and freshening occurs at the surface and the sea-

ice extent moves southward (Figure 6a,b,d,e). This produces a decrease of the surface density over most of the convective sites resulting in convection reduction in the beginning of the perturbation period (Figure 6h) ; except east of Iceland where convection increases. We note here, that the use of a higher resolution model for the ocean might lead to a reduced amount of freshwater advected over the convective areas, and therefore to a dampened reduction of the AMOC or Northern Hemisphere surface temperatures in comparison with the results presented in Figure 5. However the latter is not very clear as other types of interaction arising in higher resolution models (such as freshwater transport by eddies) might also amplify the freshwater exchanges between the boundary currents and convective sites (e.g., Swingedouw et al., 2022).

At the end of the perturbation period, the hydrographic variables almost returns to their control values and the ice-edge is close to the one of the control simulation. Nevertheless, the end of the perturbation period presents a slight warming at the surface in the north east Atlantic and a negative salinity anomaly remains at the surface in the northern Nordic Seas, while a slightly positive salinity anomaly is visible south of the sea-ice edge (Figure 6c,f).

South of the sea-ice edge, the positive temperature and salinity anomalies are present not only at the surface but also at subsurface along the Atlantic Water pathway (Figures 6c,f and 7c,f). The positive anomalies therefore suggest a resumption of the AMOC and renewed influx of heat and salt of Atlantic origin following the period of maximum AMOC slowdown.

North of the sea-ice edge, the freshening pattern at the surface is caused by the accumulation of continental melt water through the duration of the perturbation (Figure 6f). Additionally, the weak meridional circulation at these latitudes does not promote the advection of the freshwater supply out of the area (Figure S7). The slight warming is driven from below, as suggested by temperature patterns at 300 m water depth (Figure 7c).

At 300 m depth (characteristic shelf draft), the subsurface warms in several locations following the freshwater release at the surface. These locations include east and south of Newfoundland, Baffin Bay, north of the Greenland sea and the Arctic ocean (Figure 7b). At the end of the perturbation, only the Nordic Seas warming remains significant with values above 1 $^{o}$C (Figure 7c). This area where the positive temperature anomaly has developed and amplified corresponds to the area where the ice flux is the most sustained over the perturbation period. The salinity also increases in the subsurface over the Nordic Seas with time, probably as the result of sea ice production and accumulation of saline Atlantic waters. The competing effects of temperature and salinity produce an overall density decrease at this depth that is smaller than at the surface (Figure S8). However, after 100 years, the vertical density gradient reverts from time to time due to accumulation of heat in the subsurface and the warm subsurface waters are mixed with surface waters during strong intermittent convective events (Figures S8,S9), leading to a positive surface temperature anomaly north of the sea-ice edge (Figure 6c).

To summarize, the amplification of the melt rates results in surface freshening and heat accumulation at the subsurface (around 300 m depth) in the Nordic Seas, corroborating the sustained ice discharges in this region. In the Baffin Bay, the amplification of melt rates also leads to surface freshening and subsurface warming, yet only for a few decades, which is consistent with less ice discharges over this area. The reason for these regional differences can be found in background oceanic temperatures and salinity. Both these quantities are quite small in the Baffin Bay by comparison to the same latitudes in the Nordic Seas (yellow contours Figures 7a, b). Therefore, the subsurface temperature anomaly that we apply to the ice sheets,

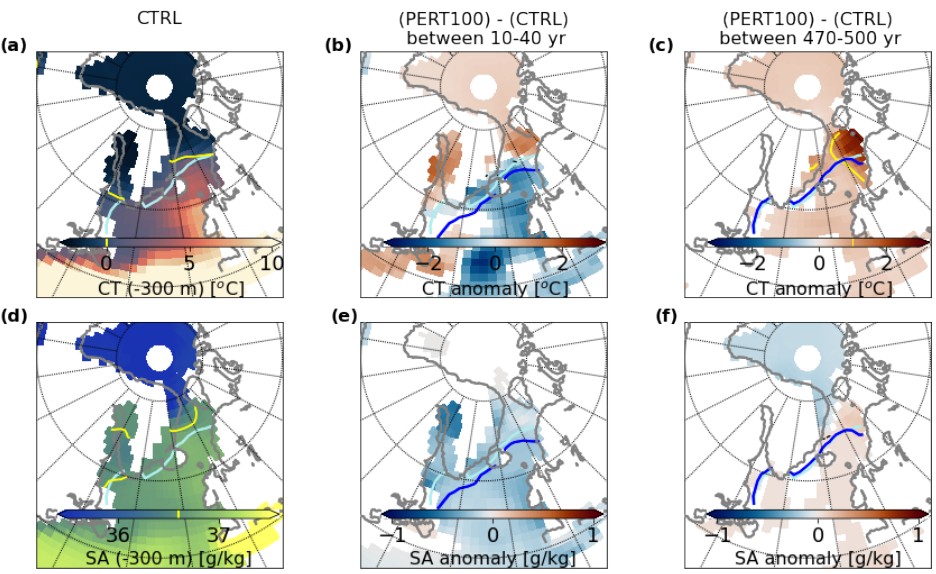

**Figure 7.** Hydrography at 300 meters depth horizon, in perturbation experiment with factor $X = 100$. Maps of (a) conservative temperature [$^oC$] over the control simulation. Conservative temperature anomalies (b) over 10-40 years and (c) over 470-500 years between the simulation with perturbation factor 100 and the control. (d,e,f) Same for absolute salinity [g/kg]. Only significant values to one standard deviation of the control simulation are shown for anomalies. Yellow contours indicate $0\ ^oC$, $+1\ ^oC$ and 36.6 g/kg iso-contours on (a), (c) and (d) respectively. Turquoise and dark blue contours are annual 30% sea ice contours in the control and the perturbation experiment respectively.

even using a large perturbation factor, may be insufficient to perturb the ice sheet dynamics there. This will be discussed in section 4.1.

### 3.3 Interactions between ice sheets and ocean

We perform another set of simulations, maintaining the oceanic perturbation and cutting off freshwater fluxes from the ice sheet model to the ocean model while the hydrological cycle of the climate model remains activated and precipitation that falls over land is still routed toward the ocean. In other words, here the ocean does not respond to the continental ice melt water and the associated oceanic feedback on the ice sheets is suppressed. Oceanic and atmospheric circulation could still vary from the control simulation, in response to changes in the ice sheet elevation (that could perturb the atmospheric physics and dynamics

for instance) or extent of continental ice (that could induce albedo changes for instance).

When the freshwater fluxes are suppressed, the ice sheet response to the perturbation is amplified during the first few decades with respect to the simulation where the freshwater fluxes are taken into account (Figures 8a,b). The Norwegian and Iceland ice sheets experience more losses (red areas; Figures 8c). In other areas such as the northern Labrador Sea (east coast of the Laurentide and west coast to the south of the Greenland ice sheets) an exacerbated thinning is also visible yet secondary.

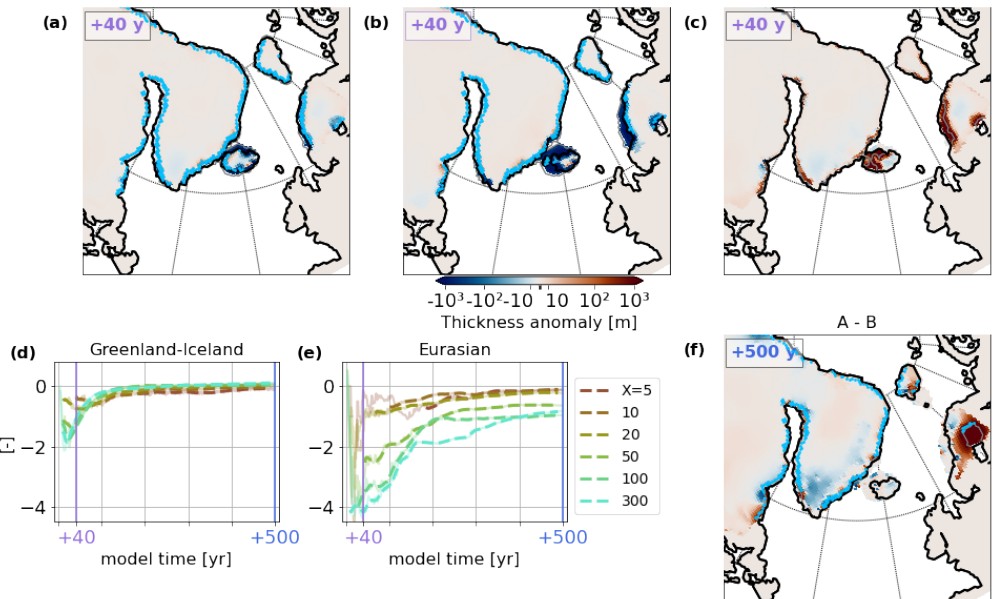

**Figure 8.** Thickness anomalies [m] between (a) perturbation experiment with factor $X = 100$ and the control simulation, (b) same with cut freshwater fluxes and (c) the difference between the two after 40 years and (f) after 500 years. Time series of (d) Greenland plus Iceland and (e) Eurasian feedback factor as defined in (Eq. 3) in the perturbation experiments. Light color lines are the yearly time series. Saturated color lines are smoothed time series (running mean over 21 years, centered) with values plotted when $\delta V_X$ is above a critical value of $0.25 \ 10^{14}$ m$^3$.

To quantify and see the temporal evolution of the freshwater feedback on the ice sheets, we compute a feedback factor $\gamma_X$ (Eq. 3).

$$\gamma_X = \frac{\delta V_X - \delta V_{X,noFWF}}{\delta V_X} \tag{3}$$

Where $\delta V_X$ is the continental ice volume variation with respect to time 0 in the simulation with perturbation factor X, $\delta V_{X,noFWF}$ in the associated simulation with no freshwater flux. This definition is adapted from Goosse et al. (2018), the 400    difference being that here we do not consider equilibriums but transient states.

For the Greenland plus Iceland and Eurasian ice sheets, $\delta V_X$ is negative (Figure 2). Thus, a negative $\gamma_X$ means that $0 > \delta V_X > \delta V_{X,noFWF}$ and indicates that there is more ice losses when the freshwater fluxes are not taken into account. Concerning the Greenland plus Iceland, the freshwater feedback on the volume variation is maximal in the beginning of the simulations (Figures 8a,b,d). After $\sim 100$ years, the ice evolution becomes stable in both experiments (Figures 2b, 8d). This is 405    because of two different mechanisms : (1) Iceland grounding line retreats inland till there is no more connection with the ocean and no more sensitivity to the perturbation, (2) the applied perturbation is not sufficient to destabilize most of the Greenland coasts, where initiated thinning only leads to reduced velocities, upstream convergence and minor thickening. The resulting difference in thickness is slightly negative in the Greenland interior at the end of the simulations (blue patch; Figure 8e). As

the thickness variation is also negative when the freshwater fluxes are taken into account, it means that there are fewer losses at this location when these are not taken into account. This is similar to what we have already described in section 3.1: the release of freshwater leads to less accumulation over the Greenland area.

Regarding the Eurasian ice sheet, the losses are amplified when the freshwater fluxes are not taken into account in the beginning of the simulation (Figure 8e). After $\sim 200$ years the time varying feedback factor becomes rather stable yet still negative for all perturbation experiments. This non-zero freshwater feedback signifies that the ice sheet volume is still decreasing in both cases, with and without freshwater fluxes, but at a larger rate for the second (Figures 2c and 8e). In both cases, the Fennoscandian ice sheet experiences a grounding line mechanical instability that implies mass loss.

Either way, ice volume decrease is faster when freshwater fluxes are not taken into account than when they are. Meltwater fluxes into the ocean bring water that is fresher than the surrounding ocean and favor sea-ice expansion, increasing stratification and reducing vertical mixing. In addition, the local latent heat flux due to oceanic melting of the calved ice also helps to cool the upper part of the water column and to maintain a cold water layer that extends from the surface to the ice shelf drafts. The latent heat flux associated with sub-shelf melt is not taken into account in the model, yet including it would lead to a larger cooling of the upper water column. All this acts to temporarily protect the ice shelves from the underlying warm waters (Figure 7c). Nevertheless, the grounding line of the Iceland and Fennoscandian ice sheets eventually retreats completely in both simulations (blue contours; Figure 6c and Figure 8f). This suggests that freshwater fluxes are slowing down the grounding line retreat and dampening the ice volume losses rather than halting them.

## 4 Discussion

### 4.1 Laurentide ice sheet

The Hudson ice stream is often considered as a major source of massive iceberg discharges during the last glacial period (e.g., Andrews and Tedesco, 1992; Bond et al., 1992; Calov et al., 2002), while the Laurentide ice sheet is overall stable in our coupled experiments. This raises the question of whether this stability is an inherent characteristic of this ice sheet simulated by the coupled model or if the disturbance applied is insufficient to cause destabilization.

On the one hand, the Laurentide ice sheet in the fully coupled model may be too stable. This could stem from the basal drag formulation, parameter values, or spatial resolution. But it could also result from biases in the climate model, in the atmosphere (a warm bias could produce widespread temperate basal conditions, by thermal diffusion through the ice), or in the ocean (low thermal forcing in the Baffin Bay).

On the other hand, we have pointed out that background temperatures are rather cold at the shelf drafts and salinity is low in the Baffin Bay and adjacent Labrador Sea in comparison with the same latitudes in the eastern part of the North Atlantic. Background oceanic basal melt rates are thus weaker (around 0.003 m/yr) at the mouth of the Hudson ice stream in the control simulation than around the Fennoscandian ice sheet (around 0.3 m/yr). Therefore, in our simulations, the oceanic perturbation imposed by subsurface temperature amplification at the shelf drafts is not able to destabilize the North American ice sheet/streams even with the highest multiplicative factor.

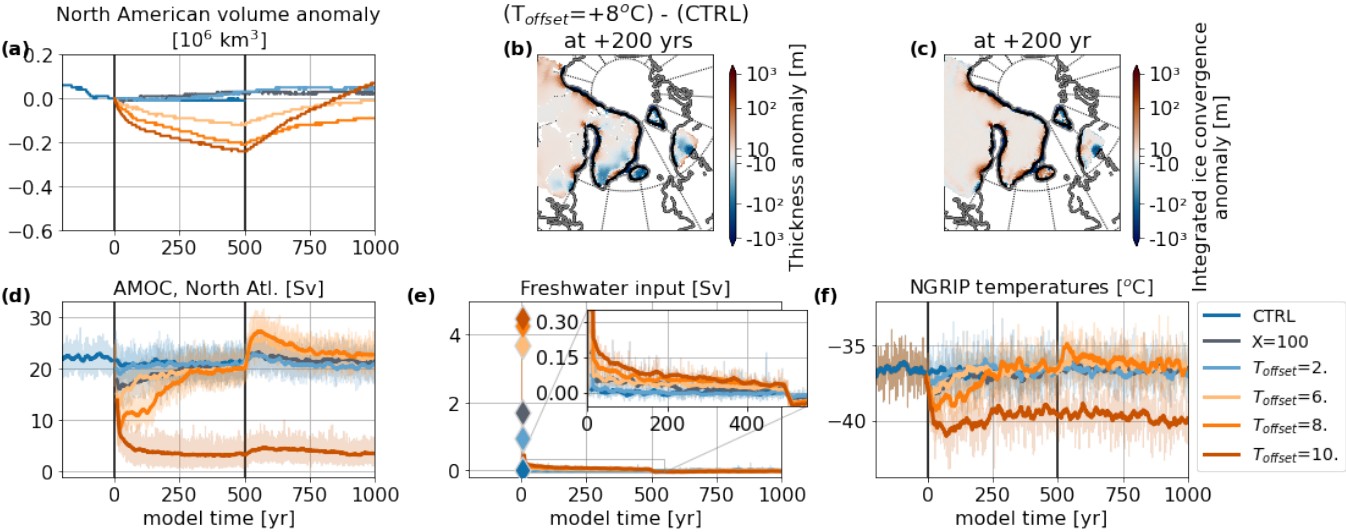

**Figure 9.** Time series of (a) North American grounded ice sheet volume anomaly [$10^6$ km$^3$], (d) AMOC [Sv], (e) Freshwater input to the ocean model [Sv] and (f) NGRIP temperatures [$^o$C] in perturbation experiments with $T_{offset} = 2, 6, 8, 10$ $^o$C as well as the experiment with X=100. Snapshot of (b) thickness anomaly [m] and (c) integrated ice convergence anomaly between the ($T_{offset} = 8$ $^o$C) and the control experiment after 200 years. Only significant values to one standard deviation of the control are shown for (b).

Inaccurate representations of the ocean circulation or hydrography may contribute to underestimated ocean temperatures in the Baffin Bay area at 40 kyr BP. Such inaccuracies might result from the model low resolution (leading to the misrepresentation of the extent of the subpolar gyre and/or recirculation patterns along the west coast of Greenland for example) or from

underestimated processes, such as the sinking of brines in the Southern Ocean, which could alter the meridional overturning circulation in the Atlantic (Bouttes et al., 2010). Improving the ocean representation is a key objective for future work. Indeed, our model can simulate carbon isotopes (Bouttes et al., 2015), allowing for direct comparison with observational data from marine sediment cores. This could serve as a constraint to improve ocean representation in subsequent studies.

To overcome these two potential issues (oceanic model biases and/or weak perturbation) and to force a different ice sheet

response especially in the Hudson Bay region, we conduct an additional set of experiments using constant temperature offsets within the basal melt rate formulation, rather than amplifying the local temperature anomalies relative to the freezing point seen by the ice shelves. In other words, we apply a $\delta T$ to $T$ in Equation 1, defined by $\delta T = T_{offset}$ with $T_{offset} = 2, 6, 8$ and 10 $^oC$ for 500 years.

This time, the North American ice sheet volume deceases (Figure 9a), together with the other ice sheet volume decrease.

This results in increased freshwater input from the ice sheets to the climate model for the largest offsets in temperature in comparison with the previous set of experiments (Figure 9e). Consequently, the reduction of the AMOC is amplified, along with a more pronounced decline of the NGRIP temperatures (Figure 9d,f). In the idealized and extreme case of a temperature offset of 10 $^oC$, the AMOC shuts down, with no recovery even after the end of the perturbation. In this case, surface cooling

and freshening is so large that convection ceases in the North Atlantic Ocean and instead shifts to the North Pacific Ocean (not shown).

Nonetheless, dynamical ice losses are also confined within coastal areas in this set of experiments despite a retreat of the grounding line (e.g., Figure 9b,c). The Laurentide ice sheet does not exhibit dynamical instability or surges of the Hudson ice stream, suggesting that this area may be too stable in our model. The simulated behaviour of the Laurentide ice sheet is inconsistent with the paleo-data and the presence of ice rafted debris of Laurentide origin within the North Atlantic marine sediments (Broecker et al., 1992; Grousset et al., 2000). There are several possible reasons for this discrepancy between the simulated behavior and the observations. It could result from the presence of biases in the ice sheet basal temperatures. In our simulations, basal conditions are widespread temperate and constant over time, while it has been shown that transitions between basal states could lead to surges for the Laurentide ice sheet (Schannwell et al., 2024). Basal temperatures depend on ice thickness, geothermal heat flux and, atmospheric temperatures. Thus, varying factors like geothermal forcing, for example, could help address this issue (Hank and Tarasov, 2024). Finally, different modelling choices could lead to different responses for the Laurentide ice sheet. Modelling choices regarding the basal dynamics or geothermal heat flux affect the ability of the ice sheet to slide over bedrock and could lead to different results (e.g., Joughin et al., 2019; Sun et al., 2020; Schannwell et al., 2023). For instance, a relatively low geothermal heat flux and/or the use of a regularized Coulomb sliding law (as opposed to a Weertman-type power law) can cause a transition from a continuously flowing Hudson Strait ice stream to long periods without flow punctuated by brief surges, as shown by Hank and Tarasov (2024).

## 4.2 Influence of the bathymetry choice on ice sheet sensitivity

Till now we used a bathymetry corresponding to the Last Glacial Maximum. Here we do the conduct the same experiment but with a Pre-Industrial bathymetry, as the 40 kyr BP bathymetry lies between these two bathymetries. The initial condition is built the same way as described in section 2.2.1. All other parameters remain unchanged.

The LGM and PI bathymetries produce different ocean currents and advection patterns of heat and salt, resulting in different distributions of water masses (Figure S10). Therefore, changing the bathymetry also produces a change in the ice sheet's geometry, leading to a reduced total volume of about 22 million $km^3$ (44 mSLE). Differences in continental ice distribution include a North American ice sheet that reaches lower latitudes on its eastern side and a Fennoscandian ice sheet of smaller extent. Additionally, the PI bathymetry also leads to a reduction of the southern-east extent of the Greenland ice sheet (Figure S10).

Due to the differing geometries and ice exposure to warmer waters in the Nordic Seas and in the Labrador Sea, the regions experiencing volume losses are not identical when we amplify the basal melt rates. The Greenland ice sheet loses significant volume along its southeastern coast, particularly at the beginning of the experiment with a large perturbation factor (X=100, Figure S11). The Eurasian ice sheet still experiences substantial losses in its northern regions. The Laurentide ice sheet undergoes dynamical thinning with this geometry, but these losses are mostly confined to its southeastern region (Figure S11), where it is exposed to warmer waters (Figure S10). This produces a relatively small difference in the amount of freshwater input to the ocean following the onset of the perturbation, with respect to the experiment performed with the LGM bathymetry. The key

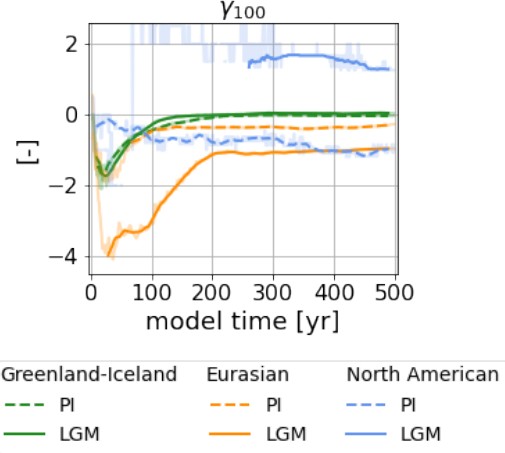

**Figure 10.** Time series of feedback factor as defined in (Eq. 3) in the perturbation experiments with factor X=100, with the Pre-industrial setup and the LGM setup, for the different ice sheets. Light color lines are the yearly time series. Saturated color lines are smoothed time series (running mean over 21 years, centered) with values plotted when $\delta V_X$ is above a critical value of $0.25 \ 10^{14} \ \text{m}^3$.

climate variables (sea-ice extent, AMOC, Northern Hemisphere and NGRIP temperatures) follow similar trajectories to those obtained with the LGM bathymetry (Figures 5,S12).

Although we could expect a change in the fully coupled model response to the oceanic perturbation following the change in bathymetry and distribution of the water masses, this is not the case here. The evolution of the feedback factor is also similar to what we obtained with the LGM bathymetry. The ice volume losses without freshwater input to the ocean are larger for both Greenland-Iceland and Eurasian ice sheets, especially at the beginning of the experiment. After $\sim 50$ years, the volume of the North American ice sheet remains stable in the experiment with the freshwater fluxes while it continues to decrease with

suppressed freshwater fluxes (Figure S13). Thus, the feedback factor is negative and the freshwater release dampens the ice volume losses with the Pre-Industrial bathymetry as with the LGM bathymetry (Figure 10). Note that $\gamma_{100}$ is positive for the North American ice sheet with LGM bathymetry, this accounts for the mass gain described in section 3.1.

### 4.3   Dansgaard-Oeschger and Heinrich events framework

Our simulations suggests that ocean subsurface warming could amplify stadial conditions. Indeed, the increase of oceanic basal

melt rates leads to a temporary increase of both calving fluxes and basal melt water release. The resulting freshwater release induces a temporary slowdown of the AMOC and decrease of the Northern Hemisphere temperature.

     According to the air temperature record of Kindler et al. (2014), the onset of Heinrich stadial 4 corresponds to $6.5 \ ^oC$ temperature decrease and its is followed by a $10 \ ^oC$ increase at the NGRIP site. In comparison, our simulations present low temperature variation magnitudes, ranging between $0.5$ and $2 \ ^oC$ for the smoothed time series (Figure 5d). In the model,

Greenland temperature changes are mainly produced by AMOC changes. Following the onset of the perturbation and the inflow of freshwater, the AMOC intensity temporarily reduces but remains relatively high (Figure 5a). Consequently, Greenland temperatures also remain relatively high, leading to small anomalies.

In addition, in the paleo-climatic records, Heinrich events are followed by an atmospheric warming over the Greenland ice sheet and North Atlantic region (Bond et al., 1993; Kindler et al., 2014), namely Dansgaard-Oeschger warming toward inter-
stadial conditions. Here, our model results do not display such warming when the perturbation is released which is inconsistent with observations (Figure 5c,d). Some authors have shown that increased sea ice extent during cold periods can lead to heat accumulation at the subsurface in the Nordic Seas (e.g., Rasmussen and Thomsen, 2004; Sadatzki et al., 2019). At some point, this accumulated heat destabilizes the water column, reversing the vertical density gradient. This reversal causes rapid oceanic mixing, bringing warm waters to the surface. The heat release to the atmosphere marks the onset of a Dansgaard-Oeschger
interstadial (Dokken et al., 2013). In our simulation, the heat accumulated in the Nordic Seas (Figure 7) dissipates once the perturbation is released (Figure S14). Part of this heat is transferred to the atmosphere, as indicated by the sea-ice contour moving toward higher latitudes, the rising ocean surface temperatures during the first 30 years post-perturbation (Figure S14) and strong, intermittent convective events during the first hundred years after the perturbation ends (Figure S9). However, this has no imprint on Greenland temperatures. One possibility is that the heat reservoir is insufficient to counteract the rapid re-
stratification that follows the surface heat release to the atmosphere (Figure S8), preventing the climate system from shifting to a new regime in our simulations. This could be explored in future studies.

The atmospheric circulation was not a primary focus of this study, yet biases in the atmosphere model could also influence the ice sheet's volume dynamics and could be invoked to explain the absence of a DO-like event following the release of the perturbation. Climate biases have been assessed for the pre-industrial climate by Quiquet et al. (2021b) : there is an overall cold
bias around the Nordic Seas sector, Greenland and Fennoscandian ice sheets, with underestimated precipitation. In contrast, there is a relatively warm bias over North America, with underestimated precipitation except in mountainous areas where it tends to be overestimated (Quiquet et al., 2021b). While it is unclear how these biases are translated at 40 kyr BP, they could affect the ice sheet's ability for regrowth and modulate the rate of its volume decay. It is noteworthy that previous studies using the iLOVECLIM model have shown its ability to simulate abrupt temperature increases at the NGRIP site ($\sim +5^oC$)
associated with a significant rise in AMOC strength ($\sim +15$ Sv) (Quiquet et al., 2021b, Figures 3 and 6). This suggests that, in our study, it is rather a lack of AMOC variability that prevents sudden warming after the perturbation is halted.

We thus see that our simulations lack some of the key features of the paleoclimatic records. Beyond the absence of Laurentide ice sheet instabilities discussed in Section 4.1, these shortcomings include an overly weak intensity of temperature anomalies at the NGRIP site and the absence of rapid warming toward interstadial conditions. These limitations are probably related to
the weak changes (in both intensity and duration) of the AMOC in response to freshwater flux originating from the ice sheets, this flux being likely underestimated over time due to the stability of the Laurentide ice sheet in our simulations.

## 5 Conclusions

In this study, we test the hypothesis that ocean subsurface warming triggers instabilities in Northern Hemisphere ice sheets. We use an Earth model of intermediate complexity (iLOVECLIM) coupled with an ice sheet model (GRISLI) and apply an oceanic perturbation at the shelves draft by multiplying the background oceanic basal melt rates with different factors. Our numerical experiments lead to several conclusions:

(1) The imposed amplified oceanic basal melt rates induce ice volume changes. Fresh water is released to the ocean through increased calving and oceanic basal melt water discharges. For most places the increase in oceanic basal melt rates leads to grounding line retreat, dynamical thinning and volume loss, while in some places the initial ice thickness decrease is small and leads to upstream minor dynamical thickening. Nevertheless, the influx of fresh water to the ocean induces a temporary increase in the sea-ice extent, a reduction in convection in the Labrador and Nordic Seas, and hence a reduction in the Atlantic meridional overturning circulation and surface temperatures in the Northern Hemisphere as well as subsurface warming in the Nordic Seas.

(2) Fresh water release and latent heat effect on ocean temperature dampen ice sheet discharges (negative feedback).

(3) The Laurentide ice sheet volume does not vary and the Hudson Strait ice stream is stable in our experiments with amplified basal melt rates. This can be partly explained by weaker background temperature and salinity at shelf drafts in the Baffin Bay and Labrador Sea in comparison with the Nordic Seas. Imposing constant temperature offsets, up to $10\ ^oC$, produces a grounding line retreat and a volume decrease of the North American ice sheet, although the Hudson Strait region shows no dynamical instability. This suggests that the model is too stable in this region.

(4) An episode of subsurface warming could trigger dynamical instabilities and ice discharges along the coasts of the Nordic Seas, affecting the Fennoscandian, Svalbard, and Iceland ice sheets. These events could lead to significant, albeit temporary, perturbation of the large-scale climate, including sea-ice expansion, a weakening of the Atlantic Meridional Overturning Circulation, and Northern Hemisphere cooling.

While our modelled Laurentide ice sheet does not exhibit dynamical instability, this does not rule out the ocean as a potential trigger for Heinrich events. Instead, the limited inland propagation of oceanic perturbations would suggest an overall stability and/or a lack of nonlinear dynamics at the ice sheet–bedrock interface in our simulations. Different modelling choices regarding the basal dynamics beneath the ice sheet (i.e. a change of the sliding law) could help to address this issue, as we know, from the presence of ice rafted debris of Laurentide origin within the North Atlantic marine sediments (e.g., Broecker et al., 1992; Grousset et al., 2000), that this ice sheet has been dynamically unstable in the past. This could be investigated in future work. Moreover, to better contextualize this study within the framework of paleo-observations of massive iceberg discharges and

Dansgaard-Oeschger events, it is also key to represent the rapid warming toward interstadial conditions following Heinrich events. The effects of the Antarctic ice sheet, which was prescribed in this study, could be examined. Indeed, the extent of the Antarctic ice sheet may also fluctuate, leading to variations in the production of deep water around Antarctica (Paillard and Labeyriet, 1994). Accounting for these variations might result in more variability in the oceanic component than the present

study. This is a potential area for future research. Lastly, our model can simulate ocean carbon isotopes (Bouttes et al., 2015), allowing direct comparison with observational data from marine sediment cores and providing additional constraints for the ocean component. This is another promising research direction.

*Data availability.* The source data of the figures in the main text of the manuscript are accessible on the Zenodo repository with the digital object identifier https://doi.org/10.5281/zenodo.12793237 (Abot et al., 2024). All color schemes are colour-vision deficiency friendly and

perceptually-uniform, accessible from a freely available package (Crameri, 2023).

*Author contributions.* LA carried out the simulations, analysed the results and wrote the paper. All other authors contributed to designing the project, the simulations, analysing the results and adding comments to improve the paper.

*Competing interests.* The contact author has declared that none of the authors has any competing interests.

*Acknowledgements.* We gratefully thank Casimir de Lavergne and Nathaelle Bouttes for their help and for early discussions. We deeply ac-

knowledge Matteo Willeit and an anonymous reviewer whose comments contributed to improving the manuscript. This work is a contribution to the French INSU (LEFE) project LASSO. LA acknowledges funding from a Sorbonne Université Ph.D. scholarship.

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
