# Peer review of "Northern Hemisphere ice sheets and ocean interactions during the last glacial period in a coupled ice sheet-climate model"

_Climate of the Past, 2024_

## Author Response (AR2)

**Editor response #2**

Though reviewer points were generally addressed, a major concern of mine given in the instructions and my prior comments has not be adequately addressed:

######
**This also entails more clearly communicating ILOVEclim strengths and weaknesses, what aspects of the experimental results are most confident and why. This needs to be addressed with consideration of all the input, parametric, and structural sources of uncertainty in**
your coupled model.

**Is CLIO still subject to flux corrections? If so, how is ISM/LoveClim # freshwater coupling implemented to avoid the flux correction offsetting the ice sheet freshwater?**
The CLIO model is indeed subject to flux corrections but the freshwater fluxes from the ice sheet to the ocean model are not.
**If this involves extensive discussion that is provided in another paper, do cite that**
for details, but all of the above do need some basic discussion in this submission as
well.
######
Following your comments on the freshwater coupling of July 05, we are sorry to have omitted to address this specific point, indeed we had only added a general content on the description of the freshwater coupling in the Supplementary S2 (at the time of the submission of July 30, 2024).

Most notably, I can find no mention of the flux corrections applied to CLIO anywhere in the revised main text or supplement (no hit with searching for 'flux c'). Given the focus on FW coupling impacts, this aspect needs to be at least mentioned in the main text and fully described in the supplement.
Are the flux corrections applied uniformly, regionally, or ? How might this affect the response to FW forcing (even if that forcing is not flux corrected)? Please provide clear reasoning as to why your results are unlikely to be biased by the flux corrections (or if you can't, describe how the flux corrections might bias the results).
Thanks for pointing this out. It is true that we overlooked this specific comment since we did not include any details about ECBilt-CLIO flux corrections. The core of the coupled ECBilt-CLIO model has been untouched from the Goosse et al. (2010) paper. We have added information in the following piece of information while describing the model in the main text:
"*Given that precipitation biases can alter the performance of CLIO to reproduce large-scale oceanic circulation, our model includes a flux correction of the precipitation rates since Goosse et al. (2010). Specifically, precipitation is enhanced over the North Pacific while reduced over the Arctic and Atlantic ocean (Supplementary S2).*"

As well as the following information in Supplementary Material S2 :
"*As precipitation is overestimated in the Atlantic and Arctic and underestimated in the North Pacific Ocean, a flux correction is applied to the precipitation rates in ECbilt before transmission to CLIO [1]. Hence, the rates are reduced by 8.5% and 25% in the Atlantic and Arctic Oceans, respectively and the excess is transferred to the North Pacific Ocean.*

*The flux corrections are applied uniformly over the basins. First, the total volume of water to be subtracted over the grid points covering the North Atlantic and Arctic Oceans is computed*

*according to the correction rates. Then, a water flux per square meter is then computed by dividing the total subtracted volume by the North Pacific Ocean surface. Finally, this additional flux is applied uniformly to the North Pacific Ocean.*

*It is difficult to assess to what extent this correction affects the results, as its removal would produce a different mean state. The Arctic Ocean (where the correction is -25%) would probably be fresher at the surface, the sea ice would be thicker and its extent would reach lower latitudes, so this would probably require some model parameters to be re-tuned. Regarding the North Atlantic Ocean (where the correction is -8.5%), we hypothesize that removing the correction would produce a slightly fresher surface layer, perhaps reducing vertical mixing in some areas. This could lead to a slightly diminished response of convection depth variability following the release of fresh water from the ice sheets. We recall here that the freshwater flux from the ice sheet to the ocean is not subject to flux correction.*"

More generally, the only discussion I could find of CLIO limitations is: "Inaccurate representations of the ocean circulation or hydrography may contribute to underestimated ocean temperatures in the Baffin Bay area at 40 kyr BP. Such inaccuracies might result from the model low resolution (leading to the misrepresentation of recirculation patterns along the west coast of Greenland for example) or from underestimated processes, such as the sinking of brines in the Southern Ocean, which could alter the meridional overturning circulation in the Atlantic (Bouttes et al., 2010)."

A 3 degree ocean model is not going to resolve boundary currents and is going to need strong diffusive mixing. Does it have a well defined sub-polar gyre (a key component identified in some previous higher resolution modelling studies examining ocean response to FWF) ? What does this all mean wrt your results?
We agree with this consideration, the purpose of this sentence was to recall that the resolution of the ocean model is rather coarse. However, despite its coarse resolution, CLIO is able to reproduce the large-scale features such as the main gyres within the North Atlantic (North Atlantic and the Subpolar Gyres), and the main currents (**Figure R1**).

[Figure]

Figure R1: Ocean velocities at the 5m depth level horizon (a) in the control simulation and (b) at the beginning of the experiment with X=100. Background colors on (b) are the speed anomalies

between the two simulations, only significant values to one standard deviation are shown. GS : Gulf stream, NAC : North Atlantic Current, IC: Irminger current, EGC : East Greenland current. Thin black contour is the modern 0m height above sea-level. Blue and green contours are respectively the 80 and 30% sea-ice contours.

The subpolar gyre delimitation, along with the mean state currents, would surely be improved with a higher resolution ocean component. The different currents forming the subpolar gyre, such as the North Atlantic Current, the Irminger Current, and East Greenland Current, would be narrower and stronger with an increasing resolution. In this sense, there will probably be less freshwater advected from the ice sheets to the convective sites resulting from the perturbation in a higher resolution model in comparison with our results. However, it is not clear how the freshwater release will really impact the convection reduction or subsequent AMOC weakening with an increased resolution. For instance, following an increased Greenland ice sheet melting, Swingedouw et al., (2022) suggested that the AMOC reduction with a high resolution ocean model (1/24°) would be larger than that of the CMIP6 models through increased lateral exchanges and freshwater transportation by eddies from the boundary currents to the convection areas.

We will add the following sentences to complement the description of the CLIO component in the main text and add the **Figure R1** in the supplement:
*"Despite its coarse resolution, CLIO is able to represent large-scale features. Taking the example of the North Atlantic Ocean, these features include the main gyres (North Atlantic and subpolar gyres), as well as the main currents. The Gulf Stream, the North Atlantic Current, the Irminger Current and the East Greenland Current are depicted, although the latter are wider than in higher resolution models (Supplementary S1)."*

Further down, in section 3.2 and in the discussion, we will add the following comments in bold to the main text :
*"This produces a decrease of the surface density over most of the convective sites resulting in convection reduction in the beginning of the perturbation period (Figure 6h) ; except east of Iceland where convection increases.* **We note here, that the use of a higher resolution model for the ocean might lead to a reduced amount of freshwater advected over the convective areas, and therefore to a dampened reduction of the AMOC or Northern Hemisphere surface temperatures in comparison with the results presented in Figure 5. However the latter is not very clear as other types of interaction arising in higher resolution models (such as freshwater transport by eddies) might also amplify the freshwater exchanges between the boundary currents and convective sites (e.g., Swingedouw et al., 2022).**"

*"Inaccurate representations of the ocean circulation or hydrography may contribute to underestimated ocean temperatures in the Baffin Bay area at 40 kyr BP. Such inaccuracies might result from the model low resolution (leading to the misrepresentation of* **the extent of the subpolar gyre and/or** *recirculation patterns along the west coast of Greenland for example) or from underestimated processes, such as the sinking of brines in the Southern Ocean, which could alter the meridional overturning circulation in the Atlantic (Bouttes et al., 2010)."*

In the end, most readers will be interested in what your results mean for understanding the actual physical system, not for understanding what LOVECLIM-GRISLI does. There needs to be more in the submission to make that interpretation possible for those who are not experts on LOVECLIM and EMICS.

Thank you for this remark, we will add the following point to the conclusion to highlight the meaning of the results:

"*(4) An episode of subsurface warming could trigger dynamical instabilities and ice discharges along the coasts of the Nordic Seas, affecting the Fennoscandian, Svalbard, and Iceland ice sheets. These events could lead to significant, albeit temporary, perturbation of the large-scale climate, including sea-ice expansion, a weakening of the Atlantic Meridional Overturning Circulation, and Northern Hemisphere cooling.*"

We will also highlight this point in the abstract, by adding the comment in bold:

"***In summary, this study suggests that an episode of subsurface warming may trigger dynamical instabilities and ice discharges along the coasts of the Nordic Seas but subsequent ocean-ice sheet interactions may be characterized by negative feedbacks, thus dampening ice discharges.*** *This study also emphasizes the need for further research using fully coupled models to explore the triggering mechanisms of massive iceberg discharges and to clarify the role of the ocean in these events.*"

We will also add intermediate conclusions at the end of each subsection, allowing for non-experts on EMICS to get the physical meaning of the main results. While this job was already done for sections 2.2.1, 2.2.2, 3.3, 4.2 lasts paragraphs, we will add the following sentences in bold at the end of section 3.1, 3.2 and 4.3 respectively :

"***At this point, this study suggests that an episode of subsurface warming may have triggered dynamical instabilities for the Eurasian, Svalbard and Iceland ice sheets. However, after about a decade, the simulated freshwater release to the ocean decreases with time.*** *The freshwater release has the ability to modify climate, ocean hydrography and circulation which can in turn affect the ice sheets. In the following section, we investigate climate and ocean changes and examine how they are connected with regional ice sheet changes.*"

"***To summarize, the amplification of the melt rates results in surface freshening and heat accumulation at the subsurface (around 300 m depth) in the Nordic Seas, corroborating the sustained ice discharges in this region. In the Baffin Bay, the amplification of melt rates also leads to surface freshening and subsurface warming, yet only for a few decades, which is consistent with less ice discharges over this area.*** *The reason for these regional differences can be found in background oceanic temperatures and salinity. Both these quantities are quite small in the Baffin Bay by comparison to the same latitudes in the Nordic Seas (yellow contours Figures 7a, b). Therefore, the subsurface temperature anomaly that we apply to the ice sheets, even using a large perturbation factor, may be insufficient to perturb the ice sheet dynamics there.* ***This will be discussed in section 4.1.***"

"***We thus see that our simulations lack some of the key features of the paleoclimatic records. Beyond the absence of Laurentide ice sheet instabilities discussed in Section 4.1, these shortcomings include an overly weak intensity of temperature anomalies at the NGRIP site and the absence of rapid warming toward interstadial conditions. These limitations are probably related to the weak changes (in both intensity and duration) of the AMOC in response to freshwater flux originating from the ice sheets, this flux being likely underestimated over time due to the stability of the Laurentide ice sheet in our simulations.***"

To conclude section 4.1, we will move the paragraph from section 4.3 about explanations regarding discrepancies between simulated behaviour of the Laurentide ice sheet and paleo-data. This will be more coherent together.

I hope that the updated submission will meet the scientific standards for publication in Climate of the Past. Thank you for your consideration,

Sincerely yours,

Louise Abot, on behalf of all co-authors

**References :**
Goosse, Hugues, et al. "Description of the Earth system model of intermediate complexity LOVECLIM version 1.2." *Geoscientific Model Development* 3.2 (2010): 603-633.

Swingedouw, Didier, et al. "AMOC recent and future trends: a crucial role for oceanic resolution and Greenland melting?." *Frontiers in Climate* 4 (2022): 838310.

**Editor report #1**

The reviewers have kindly provided detailed and thoughtful assessments of this submission. Though they are generally favourable about the ultimate value of this submission, they both recommend major revisions. A common theme was need for more clarity on the relationship between the experiments and past earth system history, as phrased by reviewer 1 " how the experiments are related to the real world". This also entails more clearly communicating ILOVEclim strengths and weaknesses, what aspects of the experimental results are most confident and why. This needs to be addressed with consideration of all the input, parametric, and structural sources of uncertainty in your coupled model. In reading your initial replies to the reviewers, I find this aspect would strongly benefit from more consideration.

A somewhat related issue is imprecise statements that then lack sufficient analysis of implications (cf example below concerning Northern Hemispheric cooling).

There are also a number of responses to the reviewer points that are not accompanied by indicated changes to the text, eg "L. 82: Does the model resolve the diurnal cycle?
No, the four hour time step in the atmospheric model is related to numerical stability."

You need to either provide a clear reason for not addressing the issue in the text or else provide the associated text change/addition (or simply state reviewer suggestion implemented when appropriate). Often this alone is the preferred whole response to a reviewer point.

Therefore please provide a more complete response along with the associated revised submission and tracked changes for re-review.
Thank you. Please find below a detailed response to your comments that will be submitted together with the associated revised version and tracked changes.

**a few specific issues in the response to reviewers**

more on the point raised by reviewer #2:
(c) Line 117: 'corresponds rather well to the extent of the reconstructed ice sheet.'
You should also consider the DATED reconstruction (Hughes et al 2015, Boreas, DOI 10.1111/bor.12142)
Thank you for the reference. DATED-1 does not provide reconstructions for periods prior to 38 kyr BP. From a temporal point of view, the closest DATED-1 reconstruction to our study is the map for the period 38-34 kyr BP. During this period, the extent of the Scandinavian ice sheet, in both the maximum and minimum DATED-1 scenarios, is less than that simulated by the fully coupled model. In the minimum DATED-1 scenario, the Scandinavian ice sheet is restricted to the mountainous areas. In the maximum DATED-1 scenario, the Scandinavian ice sheet is larger yet not marine terminating. Also, there is no Svalbard ice sheet in the DATED-1 reconstruction.
We will mention this work by adding the two following sentences : "*The DATED-1 reconstruction for the Eurasian ice sheet (Hughes et al., 2015) presents a reduced extent of the Fennoscandian ice sheet and no Svalbard ice sheet compared to our simulated ice sheet. However, this reconstruction corresponds to the 38–34 kyr BP period, which is slightly later than the time slice examined in this study.*"

"In the ice sheet model, the floating ice shelves are cut and calved according to the following procedure. The cutting thickness hcut .."

Please ensure to not just describe but also motivate/justify this treatment.

Thank you. We will complement the description of the calving law in the revised version of the Supplementary Material's section, by adding the following sentences in bold :

"***In the ice sheet model, the calving law is simple and is based on a minimum cutoff threshold. It is easy to implement and it ensures that the ice shelves do not become thinner than a minimal value (hereafter referred to as hcutpla). Indeed, thin ice shelves are more prone to breakage due to processes that are not represented in the model, such as ice shelf rifting. The cutting threshold for the ice shelf thickness above the continental plateau, hcutpla, is set to 250 m, which reflects a typical value of the observed ice shelf thickness near the ice shelf front in Antarctica todat.***

*The floating ice shelves are cut and calved according to the following procedure. The cutting thickness hcut is a function of the bathymetry d and four parameters : dpla ('depth of the plateau'), daby ('depth of the abysses'), hcutpla (minimum ice shelf thickness above the plateau), hcutdab (minimum ice shelf thickness above the plateau). Between, dpla and daby, hcut is a linear function such that hcut(dpla)=hcutpla and hcut(daby)=hcutdab.*

*For every grid points :*
*if there is an ice shelf with thickness H and the bathymetry is d :*
*dH = elevation change due to the upstream flow, surface mass balance and basal melt*
*if H+dH < hcut(d) then the ice shelf is calved*

***A critical region in our experiment is the Baffin Bay where ice shelves can easily develop due to positive surface mass balance, low sub-shelf melt rate due to low oceanic temperatures and ice convergence fed by Greenland and North American ice streams. In this region the model tends to build-up thick ice shelves that end up grounded on the seafloor. Once grounded, the ice there becomes insensitive to oceanic temperatures and remains present during the entire length of the simulation. Since this is unlikely to have happened in the past, we prevent this in our configuration. In our setup, dpla = 250 m and daby = 1500 m, hcutpla = 250 m and hcutaby = 2000 m. The minimal thickness choice of 2000 m for hcutaby prevents the shelves from developing over the deep ocean.***"

"When dV is negative then freshwater is routed toward the ocean, such as an exchange between the ice sheet and the ocean model."
Ensure you also explain the treatment for dV >0.

Thank you. We will rephrase the description of the freshwater coupling between the ice sheet and the ocean in the corresponding Supplementary Material section :
*"When dV is negative (resp. positive), freshwater is routed towards (resp. taken from) the ocean, thereby representing a freshwater exchange between the ice sheet and the ocean model."*

**a few specific issues in the manuscript**

"It includes a thermodynamic sea-ice component" Correct terminology is thermodynamic-dynamic unless you have turned off the sea ice dynamics (transports from wind stress and ocean currents).

Thank you. We will modify the sentence using "*thermodynamic-dynamic*" instead of "*thermodynamic*".

"Our study shows that amplified oceanic basal melt rates lead to  significant freshwater release from both increased calving and basal melt fluxes. "
I would suggest framing this a bit differently given that your "study is showing" a result that would largely be expected from first principles.

Thank you. We will rephrase this sentence of the abstract such as : "*The amplification of oceanic basal melt rates leads to significant freshwater release from both increased calving and basal melt fluxes.*"

"These are downscaled to GRISLI's resolution (Quiquet et al., 2018):"
Most readers are not going to bounce to the cited references to ascertain details. Therefore, provision of more details on the non-trivial downscaling employed would be beneficial, and requires only an added phrase or sentence.

In the revised version of the manuscript that we uploaded on July 30, following your original editorial comments, the corresponding paragraph had been modified into :

"*In iLOVECLIM, the atmospheric model ECBilt includes an online downscaling (Quiquet et al., 2018). At each atmospheric time step, temperature and precipitation are computed on the finer resolution GRISLI grid using the standard ECBilt energy and moisture equations. This subgrid information is fed into an Insolation Temperature Melt (ITM, Pollard, 1980; Van Den Berg et al., 2008) model to compute surface mass balance every four hours.*"

We believe that this paragraph takes your comment into account and suppose that you have not seen our first revised version uploaded on July 30.

"lowers surface temperatures in the Northern Hemisphere"
If this is the case, then terrestrial ice sheet area should expand. But there is no discussion as to whether this is the case. Or do you mean surface temperatures only lower for certain Northern Hemispheric regions away from terrestrial margins?

Thank you for this remark.

The Northern Hemisphere cooling is mostly driven by surface temperature decrease in the North Atlantic when the perturbation is triggered (**Figure R1** below). Indeed, the surface cools locally for certain Northern Hemispheric regions. The largest cooling occurs in areas such as the Nordic Seas or Labrador Seas, minor cooling occurs over Greenland and Europe, and there is no significant (to 1 standard deviation of the control experiment) temperature change over North America.

As the cooling occurs away from the continental margins, the ice sheet area does not expand.

Besides, the Northern Hemisphere cooling is only a transient response to the freshwater discharges (through latent heat effect on ocean temperature, increased sea ice extent and reduced convection). It is possible that this response does not last long enough for the ice sheet to 'see it'.

[Figure]

Figure R1: Surface temperature [°C] in (a) the CTRL experiment, and associated anomalies between the perturbation experiment with X=100 (b) between the first 50 years and (c) the last 50 years. Only significant values to one standard deviation of the CTRL experiment are shown.

We will add the following comment in Section 3.1 : "*Although one could expect that the transient Northern Hemisphere cooling would allow ice sheet regrowth during the perturbation period, this is not the case. Indeed, the Northern Hemisphere surface cooling decrease is essentially located over the North Atlantic Ocean (especially over the Nordic and Labrador Seas), while only limited cooling occurs over Greenland and Europe, and no notable temperature changes take place over North America.*"

"The basal drag is directly proportional to the basal velocities over the grounded ice sheet"
If you mean you are using linear Weertman sliding, then please state so. "Directly proportional" does not necessarily mean linear.
Thank you for this comment. This sentence will be replaced by the following :
"*When the ice sheet's base is at the pressure melting point, the basal drag is a linear function of the basal velocities over the grounded ice sheet (Weertman, 1957).*"

Loveclim has numerous poorly constrained parameters. Are you using the default values from the Loveclim source archive? If so, please clearly state so. If not, please provide values used.
Thank you for this remark. Yes, as described in the supplementary material S1 of the revised manuscript that we submitted at the end of July after having taken your initial remarks into account, we use the default values from the iLOVECLIM source archive, except for the Bering Strait scaling factor and climate sensitivity factor.

We will complement this point by adding the sentences in bold to the main text : "*Major model parameters are listed in Supplementary Tables S1a-c.* **We use the default values from the iLOVECLIM source archive, except for the Bering Strait scaling factor and climate sensitivity factor. The first is set to zero to ensure a closed Bering Strait. The second is artificially doubled (Timm and Timmerman, 2007) to achieve a relatively cool climate and maintain the ice sheets at 40 kyr BP. This modelling choice is informed by the relatively low equilibrium climate sensitivity of iLOVECLIM, which is approximately 2°C (Bouttes et al., 2024), in comparison with the IPCC AR6 best estimate of 3 °C and likely range of 2.5 to 4 °C (IPCC, 2023).**"

The model parameters have been added in Table S1a and S1b of the Supplementary Material (**Figure R2** below), following your original editor comments. The same has been done for GRISLI parameters in Table S1c. These tables are referred to in the main text.

**Table S1a :** Main parameters of the ECBILT module (atmosphere component).

| Parameter | Definition | This study *(default)* |
|---|---|---|
| ampwir | Scaling coefficient for the longwave radiation scheme, excluding the equator area | 1.0 |
| ampeqir | Same, for the equator area between $15^oS$ and $15^oN$ | 1.8 |
| expir | Exponent for the longwave radiation scheme | 0.4 |
| relhmax | maximum relative humidity before saturation | 0.83 |
| hmoisr | reduction factor of mountain heights in order to tune the amount of water that is allowed to pass a topographic barrier | 0.5 |
| umoisr | reduction factor of 800 hPa winds used in advecting the moisture | 0.6 |
| rrdef1 | Rossby radius of deformation layer 1 (200 - 500 hPa) | 0.13 |
| rrdef2 | Rossby radius of deformation layer 2 (500 - 800 hPa) | 0.07 |
| cwdrag | drag coefficient to compute wind stress | $2.1 * 10^{-3}$ |
| cdrag | same to compute sensible and latent heat fluxes | $1.4 * 10^{-3}$ |
| uv10rfx | reduction of the wind speed between 800 hPa and 10 m | 0.8 |
| dragan | rotation of the wind vector in the boundary layer | 15.0 |
| alphd | albedo of snow | 0.72 |
| alphdi | albedo of bare ice | 0.62 |
| alphs | albedo of melting snow | 0.53 |
| albice | albedo of melting ice (general) | 0.44 |
| albin | albedo of melting ice (arctic) | 0.44 |
| cgren | increase in snow/ice albedo under cloudy conditions | 0.04 |
| corAN | reduction in precipitation over the North Atlantic | -0.085 |
| corPN | reduction in precipitation over the North Pacific | 1.0 |
| corAS | reduction in precipitation over the South Atlantic | -0.085 |
| corAC | reduction in precipitation over the Arctic | -0.25 |
| mag_alpha | coefficient for the climate sensitivity | 2.0 *(1.0)* |
| evfac | maximum evaportation factor over land | 1.0 |

**Table S1b :** Main parameters of the CLIO module (ocean component).

| Parameter | Definition | This study *(default)* |
|---|---|---|
| bering | Scaling factor for computing the Bering Strait throughflow | 0.0 *(0.3)* |
| ai | Coefficient of isopycnal diffusion | 300 |
| aitd | Gent-McVilliams thickness diffusion coefficient | 300 |
| ahs | Horizontal diffusivity for scalars $[m^2 s^{-1}]$ | 100 |
| ahu | Horizontal viscosity $[m^2 s^{-1}]$ | $10^5$ |

Figure R2: Supplementary Tables S1a and S1b.

"In our setup, the iLOVECLIM-GRISLI coupled model, under constant or transient forcings corresponding to the Last Glacial Period, does not exhibit abrupt internal oscillations, neither in the ice sheets nor in the ocean circulation".

"This simulated behaviour of the Laurentide ice sheet is inconsistent with the paleo-data and the presence of ice rafted debris of Laurentide origin within the North Atlantic marine sediments (Broecker et al., 1992; Grousset et al., 2000). There are several possible reasons for this contradiction:"

As I raised in my original editor comments and which has not been addressed, this may also be due other factors that you do not discuss such as choices (including numerical) made in the configuration of GRISLI along with chosen basal heat fluxes. In this regard, I again draw your attention to a recent detailed ice sheet centric examination of surge cycling controls for Hudson Strait ( Hank and Tarasov, CP : https://doi.org/10.5194/cp-20-2499-2024) as well as an earlier consideration of numerical issues (Kevin Hank, Lev Tarasov, and Elisa Mantell, GMD 2023).

Thank you for raising this. In the manuscript version that has been sent to the reviewers, after your original editor comments, we added the following sentences, line 340-342 : *"We note here that different modelling choices could lead to different responses for the Laurentide ice sheet. For instance, modelling choices affecting the basal dynamics of the ice sheet and its ability to slide over bedrock could lead to different results, as suggested by Hank and Tarasov (2024)."*

We will re-emphasize this point in the revised version, as follows :
*"Finally, different modelling choices could lead to different responses for the Laurentide ice sheet. Modelling choices regarding the basal dynamics or geothermal heat flux affect the ability of the ice sheet to slide over bedrock and could lead to different results (e.g., Joughin et al., 2019; Sun et al., 2020; Schannwell et al., 2023). For instance, a relatively low geothermal heat flux and/or the use of a regularized Coulomb sliding law (as opposed to a Weertman-type power law) can cause a transition from a continuously flowing Hudson Strait ice stream to long periods without flow punctuated by brief surges, as suggested by Hank and Tarasov (2024)."*

The GRISLI configuration needs to be more detailed as this will affect the surge cycling response of your simulated ice sheet. What is the functional relationship between basal sliding and basal temperature (ie when proximal to the pressure melting point)? What geothermal heat flux do you use? What soft-hard basal drag mask do you use? What are the basal sliding coefficients for hard and soft beds? What enhancement factor is used for the flow law? Much of this information can be succinctly communicated in a table.

Thank you for these questions. We will complement the Section 2.1.2 on GRISLI description by adding the following paragraph : *"When the ice sheet's base is at the pressure melting point, the basal drag is a linear function of the basal velocities over the grounded ice sheet (Weertman, 1957). When the ice sheet's base is below the pressure melting point, the sliding velocity is set to zero (infinite friction). The friction is set to zero for the floating ice shelves. Following the Weertman law, the basal drag is equal to the opposite of the basal sliding velocity times a basal drag coefficient β. This coefficient β is set to the effective pressure at the base of the ice sheet times a tuning parameter (Cf) and is bounded by a minimum ($β_{min}$) and a maximum ($β_{max}$) value. Over thick sediment layers (more than 200 m) the basal drag is reduced by a factor 20. We use the soft-hard basal drag mask (i.e. the sediment map) from Laske (1997) and the geothermal heat flux from Shapiro and Ritzwoller (2004)."*

The enhancement factor (Ef) coefficient is included in the Table S1c, it is set to 1.826. The Table S1-c for GRISLI parameters will be complemented with additional factors that were not already mentioned in Table S1c, specifically $\beta_{min}$ = 100 Pa.yr/m and $\beta_{max}$ = 500000 Pa.yr/m.

**Table S1c :** Main parameters of GRISLI (ice sheet model) and parameters related to the coupling.

| Parameter | Definition | This study (*default*) |
|---|---|---|
| $E_f$ | Flow enhancement factor for vertical shearing with a Glen viscosity [-] | 1.826 |
| $C_f$ | Basal drag parameter [-] | $3*10^{-4}$ ($1.6*10^{-3}$) |
| $\beta_{min}$ | Lower bound for the basal drag coefficient [Pa.yr.m$^{-1}$] | $10^2$ |
| $\beta_{max}$ | Upper bound for the basal drcag coefficient [Pa.yr.m$^{-1}$] | $5*10^5$ |
| $\lambda$ | Longwave radiation coefficient for melt computation [W.m$^{-2}$.K$^{-1}$] | 10 |
| $c_{rad}$ | Radiative parameter for melt computation [W.m$^{-2}$] | -40 |
| $\gamma_T$ | Heat transfer velocity [m/s] | $1.1*10^{-5}$ (*) |

Figure R3: Supplementary Table S1c.

**Discussion/ 4.3 Dansgaard-Oeschger and Heinrich events framework**
"Indeed, we have designed our initial conditions based on the ice sheet reconstructions from Gowan et al. (2016)."
This statement is confusing since the model has full two-way coupling between ice and climate. Furthermore, in section 2.2.1 Initial condition, there is no mention of Gowan et al. Instead you only state:
"This simulation starts from a previous equilibrium simulation of the Last Glacial Maximum (LGM) similar to the one described in Quiquet et al. (2021)."
Thank you. We agree that the latter statement was misleading, we will modified this sentence in the revised version as follows :
"*Second, the use of alternative geometries, with ice free conditions over Hudson Bay, closer to the ice sheet reconstructions from Dalton et al. (2022a, b), might produce different results.*"

"Upstream the coastal thinning, thickening takes place for the Greenland ice sheet, especially along the east coast. The latter is associated with the largest velocity decreases after 200 years (< −100 m/yr ; yellow contour on Figure 3d). In contrast, ice flow velocities increased with the grounding line inland retreat on the Fennoscandian, Svalbard and Iceland coasts"
Why the contrasting response? This is partly addressed in 3.1/3.3/4.1, but warrants more unified consideration. Many readers might wonder why increased mass-loss at a marine margin would result in upstream thickening. Eg just stating " Here, a reduction of the thickness due to the perturbation induces a reduction of the ice flux and upstream convergence " raises the question of why the ice flux decreases given that forced marginal thinning and upstream thickening will increase upstream ice surface slopes and therefore increase driving stress...
Thank you for this important question.

The contrasting responses arise from the regional differences in both initial ocean temperature and salinity (through the freezing temperature) at the shelf drafts. The regional differences in hydrographic state lead to regional differences of the initial basal met rates (as illustrated by Figure R4a,d; given in response to reviewer#1). For instance, before the triggering of the oceanic perturbation, the initial basal melt rates are of the order 0.01 m/yr for the Svalbard exposed to the Nordic Seas, 0.001 m/yr for the Greenland east coast, between 0.01 and 1 m/yr for the Norwegian coast and up to 10 m/yr for Iceland. The initial shelf drafts also follow regional patterns, and a thin

ice shelf would a priori tend to disappear quicker than a thick one, also contributing to the contrasting responses.

The thickening is only minor (<100 m) and reflects a second order effect of the oceanic perturbation (see **Figure R4**, that is the same as Figure 3 from the manuscript but without the logarithmic scale for the colorbar and saturating the extremum values for the thickness at +/- 500 m). Indeed, when the perturbation is small, the grounding line does not move and remains at the same location. Therefore, no dynamical instability arises over the corresponding ice sheet.

Failing to trigger a grounding line migration, the perturbation only induces a reduction of the ice thickness and a reduction of the ice flux in the grounding zone, following the parameterization of Tsai et al., (2015). In the latter formulation, the ice flux in the grounding zone is proportional to the ice thickness to a power n+2, with n=3, the Glen's flow law exponent. This results in positive ice convergence when ice velocities upstream of the grounding zone undergo a smaller decrease. However, the ice elevation due to ice convergence in this area is too small to lead to a discharge through an increase of the slope.

[Figure]

Figure R4: Ice sheet history in perturbation experiment with factor X = 100. Ice sheet (a) velocity [m/yr] and (b) thickness [m] averaged over the control simulation. Sky blue contour indicates the position of the grounding line. PERT100 - CTRL anomalies of (c) thickness, (d) ice velocities, (e) thickness due to climate anomalies, (f) due to ice convergence, after 200 years. Only significant values to one standard deviation of the control are shown for (c) and (d).

The main text will be revised by adding the following sentences in bold :
"*Minor thickening (< −100 m ; note that the color scale for Figure 3c,e,f is logarithmic) occurs over the Greenland ice sheet, especially along the east coast, upstream of coastal thinning. The latter is associated with the largest velocity decreases after 200 years (< −100 m/yr ; yellow contour on*

*Figure 3d).* **In this area, the perturbation fails to trigger a grounding line retreat.** *In contrast, ice flow velocities increased with the grounding line inland retreat on the Fennoscandian, Svalbard and Iceland coasts, sometimes until there is no more connection between ocean and land as it is the case for the Iceland after 200 years (blue contours ; Figures 3b versus d).* **These contrasting responses reflect regional differences in the initial basal met rates (Figure 1c) that arise from regional differences in ocean temperature and salinity (through freezing temperature) at the shelf drafts, as we shall see in section 3.2.**"

And latter in the same section :
"*The thinning due to the dynamic term is most important in areas such as the west coast and southern part of the Fennoscandian (proglacial lake grounding line instability for the latter), the Svalbard, Iceland and southern tip of the Greenland ice sheet.*
*Also, the minor thickening in coastal areas is due to the dynamic term. Indeed, in the model, the ice flux at the grounding line connects the ice thickness and velocities in the grounding zone through a power-law relationship (Tsai et al., 2015).* **Where the oceanic perturbation fails to trigger a grounding line retreat, the perturbation only causes a reduction in thickness that leads to a reduction of the ice flux and to upstream convergence (Figure 3f). However, the resulting ice elevation increase in these areas is insufficient to produce a discharge through an increase of the ice sheet surface slope.**"

Furthermore, we will insist on the fact that this thickening is small when referring to it everywhere else in the text.

Thank you for the report and all the comments,

Louise Abot, on behalf of all co-authors.

**References**

Bouttes, Nathaelle, et al. "Projections of coral reef carbonate production from a global climate-coral reef coupled model." *EGUsphere* 2024 (2024): 1-25.

Hank, Kevin, and Lev Tarasov. "The comparative role of physical system processes in Hudson Strait ice stream cycling: a comprehensive model-based test of Heinrich event hypotheses." *Climate of the Past* 20.11 (2024): 2499-2524.

Laske, Gabi. "A global digital map of sediment thickness." *Eos Trans. AGU* 78 (1997): F483.

Schannwell, Clemens, et al. "Sensitivity of Heinrich-type ice-sheet surge characteristics to boundary forcing perturbations." *Climate of the Past* 19.1 (2023): 179-198.

Shapiro, Nikolai M., and Michael H. Ritzwoller. "Inferring surface heat flux distributions guided by a global seismic model: particular application to Antarctica." *Earth and Planetary Science Letters* 223.1-2 (2004): 213-224.

Sun, Sainan, et al. "Antarctic ice sheet response to sudden and sustained ice-shelf collapse (ABUMIP)." *Journal of Glaciology* 66.260 (2020): 891-904.

Timm, Oliver, and Axel Timmermann. "Simulation of the last 21 000 years using accelerated transient boundary conditions." *Journal of Climate* 20.17 (2007): 4377-4401.

Tsai, Victor C., Andrew L. Stewart, and Andrew F. Thompson. "Marine ice-sheet profiles and stability under Coulomb basal conditions." *Journal of Glaciology* 61.226 (2015): 205-215.

Weertman, Johannes. "On the sliding of glaciers." *Journal of glaciology* 3.21 (1957): 33-38.

IPCC, 2023: Sections. In: Climate Change 2023: Synthesis Report. Contribution of Working Groups I, II and III to the Sixth Assessment Report of the Intergovernmental Panel on Climate Change [Core Writing Team, H. Lee and J. Romero (eds.)]. IPCC, Geneva, Switzerland, pp. 35-115, doi: 10.59327/IPCC/AR6-9789291691647

The answers to reviewers #1 and #2 are added below. Where they were missing, modifications of the main text have been specified by the sentences in bold.

**Detailed response to Reviewer #1's comment**

**General comments**

In the paper the authors present results from simulations of a coupled climate-ice sheet model exploring the interaction between ice sheets and the ocean during glacial times. In particular they apply different scaling factors to the basal melt rate and investigate the resulting ice sheet model response. The coupled climate model further allows them to explore the subsequent impact of the ice sheet meltwater input on the ocean state and its feedback on ice sheets. They show that scaling the basal melt rate generally results in a decrease in ice sheet volume, which is more pronounced for the Fennoscandian and Greenland/Iceland ice sheets than for the Laurentide ice sheet. The freshwater input results in an AMOC weakening, sea ice expansion and cooling and provides a negative feedback on ice sheet retreat.

The paper is generally well written and clearly organized. However, I have some comments that should be addressed before the paper is suitable for publication.
Thank you for your careful reading and the time you have spent reading and commenting on the manuscript.

**Major comments**

I'm missing some more guidance in the interpretation of the experimental setup and how the experiments are related to the real world. What are the simulations trying to represent? Is it trying to explain the climate-ice sheet response to something like DO or Heinrich events?
Thank you for these questions. Indeed, in the literature of the last decade, it has been suggested that oceanic temperature variability in the Atlantic (driven by oceanic circulation change for instance) may be a trigger for massive iceberg discharges and Heinrich events (e.g. ; Alvarez-Solas et al., 2010, 2013, Barker et al., 2015). So, the perturbation we apply aims to mimic changes in the heat supply to the ice shelves and how it could have led to massive iceberg discharges during the last glacial period (here at 40 kyr BP).

This theory has not been verified with fully coupled models that include an ice sheet component, while there could be potentially large feedbacks on ice loss arising at the interface that are due to the release of freshwater. Fully coupled models are useful to capture the feedbacks between components and the iLOVECLIM climate model coupled with the ice sheet model GRISLI is such a model.

The ocean-driven ice sheet changes might come from internal oceanic oscillations (e.g. salt oscillator). However, in our setup, we do not have self-generated oceanic oscillations in our simulations, under constant 40 kyr external forcings but also under transient 40-20 kyr forcings. This is why we designed experiments with amplified sub-shelf melt perturbations. We wanted to test how the ice sheet dynamic, in the fully coupled model, can be perturbed by the ocean, as well as the signs of the feedbacks at the ice sheet-ocean interface within the model.

We agree that the context was not well highlighted. We will emphasize the massive iceberg discharges framework in the revised manuscript by adding the two following paragraphs in the Introduction section :

*"The Last Glacial Period began around 75 kyr before present (BP) and ended with the Last Glacial Maximum (LGM) around 21 kyr BP. An intriguing feature of the last glacial period is its millennial-scale variability, first revealed in Greenland ice cores, known as the Dansgaard-Oeschger (DO) cycles (Dansgaard et al., 1984, 1993). These are transitions between relatively cold (stadial) and warm (interstadial) periods in the Northern Hemisphere (Johnsen et al., 1992). The rapid warming (or DO event) generally takes place over a period of several decades and is of the order of 10oC at the North Greenland Ice Core Project (NGRIP) site (Kindler et al., 2014). These events occurred at a frequency of around 1, 500 yr (Schulz, 2002), although this frequency is debated (Ditlevsen et al., 2007). They are followed by a slower cooling of the order of a hundred to a thousand years, and the cycle ends with a sudden cooling (Kindler et al., 2014).*

*Some of the stadials are accompanied by massive iceberg discharges from the Northern Hemisphere ice sheets in the North Atlantic Ocean, at a frequency of several thousand years (Heinrich, 1988; Bond and Lotti, 1995). The iceberg discharges are identified in marine sediment cores by one or several layers of ice rafted debris, the materials entrapped and transported by drifting ice (Bond et al., 1992; Bond and Lotti, 1995; Elliot et al., 1998). The iceberg discharges are referred to as Heinrich Events (HE) when the continental ice is originating from the Laurentide ice sheet, the eastern part of the North American ice sheet. The Heinrich events mostly occur after several DO cycles and are followed by a DO event (Bond et al., 1993). Barker et al. (2015), through the examination of a site southwest of Iceland, suggested that massive iceberg discharges were a consequence of prolonged stadial conditions. It was suggested that massive iceberg discharges occur after an initial reduction of the Atlantic Meridional Overturning Circulation (AMOC) and the build up of a heat reservoir at the subsurface in the North Atlantic from both observations and modelling studies (e.g., Jonkers et al., 2010; Max et al., 2022; Mignot et al., 2007)"*

We will also explain with more details the experimental setup and how experiments are related to the real world by adding the following explanations to the corresponding section :

*"This scaling (X) of the basal melt is equivalent to imposing a temperature anomaly, that depends on the water masses and background state of the ocean, at the shelf drafts. Specifically, we apply a perturbation whose intensity is proportional to the local temperature anomaly with respect to the freezing temperature, $T_f$. Indeed, when $T > T_f$, adding a perturbation term $\delta T$ to $T$, defined by $\delta T = \mathcal{E}(T-T_f)$ (with $\mathcal{E}>0$) leads to the following oceanic basal melt rate : $OBM=\gamma_T A(T+\mathcal{E}(T-T_f)-T_f)^2 = \alpha(1+\mathcal{E})^2(T-T_f)^2$. And we rewrite $X=(1+\mathcal{E})^2$. So there is a direct correspondence between X and the temperature anomaly intensity $\mathcal{E}$ seen by the ice shelves. For example, a doubling of the temperature anomaly with respect to the freezing point corresponds to $\mathcal{E} = 1$ and $X = 4$. A 10 time increase of the temperature anomaly corresponds to $\mathcal{E} = 9$ and $X = 100$. "*

What is the increase in basal melt intended to represent? Subsurface warming during DO stadials? Why is basal melt increased instantaneously?
Thank you for these comments. We will complement the Perturbation experiment section by adding the following sentence :

*"This way, the perturbation experiments aim to represent an increased heat flux to the ice shelf drafts, reflecting subsurface ocean warming in the North Atlantic during a Dansgaard-Oeschger (DO) stadial (e.g., Rasmussen and Thomsen, 2004)."*

We have chosen to increase the basal melt instantaneously, as this seemed to be the most direct way to evaluate the sign of the feedback and to maximize the ice sheet response to the perturbation. Proceeding differently did not seem to serve the purpose any better at this stage of the study. We will also add a sentence to comment on that :

*"Here, we amplify the basal melt rates instantaneously as this seems to be the most direct way to maximize the ice sheet response to the perturbation and to evaluate the sign of the feedbacks at the ice-ocean interface."*

However, we took good notice of this question and ran another set of experiments, with a gradual increase in basal melt rates rather than an instantaneous increase (**Figure R1**b). This way, the ice volume decrease is delayed and smoothed at the beginning (see for example the time derivatives around yr = 0 on **Figure R1**a, Greenland-Iceland or Eurasian) in comparison with the previous experiment (grey versus orange lines on **Figure R1**). Thus, less freshwater is brought from the continent toward the ocean and in a less abrupt manner. Therefore AMOC response to such a perturbation is also dampened (**Figure R1**c).

[Figure]

Figure R1 : Time series of (a) grounded ice volume [10⁶ km³] for the North American, Greenland-Iceland and Eurasian ice sheets in perturbation experiments, with the (b) perturbation factor X [-] as a function of time, and (c) AMOC [Sv] for the same experiments.

Thank you for mentioning it. We will add this to the Supplementary Material and add the following sentence to the main text:

*"Note that the impact of a gradual increase in basal melt rates on ice loss is similar to that of an instantaneous increase (Supplementary S4)."*

A map showing more clearly the ice shelf thickness in the control run would be useful to interpret the results. Similarly, the 2D field of basal melt rate would provide useful information about the spatial distribution of reference melt rate. Both ice shelf thickness and basal melt could be conveniently integrated as additional panels into Fig. 1.
That is a good idea. Thank you for this recommendation. We will change the Figure 1 of the main text for the **Figure R2** below.

[Figure]

Figure R2 : 40 kyr equilibrium. Ice sheet elevation (shades of blue) [m] at the end of the equilibrium simulation. Light blue contour is the mean position of the grounding line at the same time. Yellow contours are the minimal and maximal ice sheets extent in the reconstruction from Gowan et al. (2021), these differ only over the central North American ice sheet. Areas used to derive regional ice volumes on Figure 2 are shaded in red, green and magenta for the North American, Greenland-Iceland and Eurasian ice sheets respectively. The black lines represent topographic contours at intervals of 500 meters. (b) Ice shelf drafts [m]  and (c) basal melt rates beneath the shelves [m/yr] at the end of the equilibrium simulation.

Why is the freshwater flux increase after the application of the basal melt scaling so large initially and decrease so quickly? Is the shelf ice generally so thin that it melts away in a few years? Or is the calving process responsible for this behavior?
Thank you for these comments. Lines 176 to 182 will be replaced by the following paragraphs :

"*Still, the overall volume loss experienced by the ice sheets is equivalent to a freshwater release toward the ocean (Figure 4a). This freshwater flux is particularly large during the first few years in all experiments and results from both abrupt increased basal melt and calving fluxes in the first few years (Figure 4b,c). The initial amplification of the melt rates leads to large losses at the beginning as a basal melt flux (Figure 4b). The initial amplification of the melt rates also leads to an abrupt increase of the calving flux (Figure 4c). Indeed, some of the shelves are rather thin before the perturbation is triggered (Figure 1b), so the perturbation causes these shelves' thickness to fall below the calving threshold (Supplementary S2). Then, the calving flux drops toward the control value after 10 years while the basal melt flux slowly decreases toward the control value. This results in a freshwater flux that decreases with time (but remains significant). This decrease is partly attributed to the reduced ice volume exposed to basal melting as ice shelves thin or vanish in certain regions, such as Svalbard and Iceland. It also suggests the presence of ice sheet stabilizing mechanisms within the model, that is, a negative feedback that mitigates ice volume loss. Here, this feedback involves a decrease of the temperature at shelf drafts, which in turn reduces the basal melt rates.*"

Following your comment the Figure 4 of the main text will be replaced by the following **Figure R3**:

[Figure]

Figure R3 : Time series of (a) freshwater output from GRISLI to iLOVECLIM [Sv] (as a total ice volume change) in experiments with different perturbation factors. Diamonds are the freshwater release for the first year of perturbation, light colors are the yearly outputs and saturated colors are smoothed time series (running mean over 21 years, centered). Times series of (b) calving flux [Sv] and (c) basal melt flux [Sv], also smoothed (running mean over 21 years, centered)

We will also change units for the basal melt and calving flux to Sverdrups following one of your comments below.

On 2D maps, we can also see the effect of the perturbation once it is activated (here at yr=+1, middle column on **Figure R4**), the basal melt increases and the thinnest shelves disappear, eventually reappearing later (last column).

[Figure]

Figure R4 : (a) Ice shelf drafts [m] and (b) basal melt rates [m/yr] for perturbation experiment with factor 100 at three different times : (left column) before the perturbation is triggered, (middle) one year after (right) and after 20 years.

To what extent are the results expected to depend on the ice shelf area in the control run?
The results are expected to depend on the model configuration and the associated ice shelf area as a larger ice shelf area would produce more melt fluxes to the ocean. We will detail this further below.
And what determines how far the floating ice extends into the ocean in the model? Is it mainly controlled by calving? This should be discussed, as the initial ice shelf area is probably relevant for

the strength of the response of the model to the applied basal melt perturbation.

Yes, the extent of the ice shelves are controlled by calving and we provide in the following a detailed explanation of how this flux is calculated. This will be added to the Supplementary Material :

*"In the ice sheet model, the floating ice shelves are cut and calved according to the following procedure. The cutting thickness hcut is a function of the bathymetry d and four parameters : dpla ('depth of the plateau'), daby ('depth of the abysses'), hcutpla (criterion for the ice shelf thickness above the plateau ), hcutdab (criterion for the ice shelf thickness above the plateau).*

*For every grid point :*
    *If there is an ice shelf with thickness H and the bathymetry is d :*
      *dH := the elevation change due to the upstream flow, local surface mass balance and basal melt.*
      *if H+dH < hcut(d) then the ice shelf is calved.*

*Between, dpla and daby, hcut is a linear function such as hcut(dpla)=hcutpla and hcut(daby)=hcutdab."*

[Figure]

```
!
!  hcut(d)
!    ^                           ______ hcutaby
!    |                      ./
!    |                   ../
!    |                ../
!    |              ../
!    | hcutpla _____/
!    -----------------^---------^------------------------ > section running from the coast to the open
!                  dpla     daby                          ocean with bathymetry d
!
```

*"A critical region in our experiment is the Baffin Bay where ice shelves can easily develop due to positive surface mass balance, low sub-shelf melt rate due to low oceanic temperatures and ice convergence fed by Greenland and North American ice streams. In this region the model tends to build-up thick ice shelves that finally end up grounded at the seafloor. Once grounded the ice there becomes insensitive to oceanic temperatures and remains present for all the simulations. Since this configuration is unlikely to have happened in the past, we prevent this in our reference configuration. In our setup, dpla = 250 m and daby = 1500 m, hcutpla = 250 m and hcutaby = 2000 m. The minimal thickness choice of 2000 m for hcutaby prevents the shelves from developing over the deep ocean."*

Nonetheless, we made new experiments that favour larger ice shelves (but still ensuring a non-grounded Baffin Bay) by setting dpla to 1000 m and daby to 2500 m (**Figure R5**) in order to assess the impact of the shelf area.

[Figure]

Figure R5 : Comparison of the last hundred years of two long spin up configurations : the one from the manuscript (REF) and a second configuration with larger shelves (NEW).

Applying the perturbation with X=100 on this configuration with larger shelves leads to similar results as before. Larger shelves melting lead to more freshwater input from the ice sheet model to the climate model, especially in the beginning of the simulations (**Figure R6**e). This is associated with a bit more reduction of the AMOC and NGRIP temperatures in comparison with the previous experiment.

[Figure]

Figure R6 : Comparison between the same perturbation experiments with factor X=100 branched on the long spin up configuration (REF) and on a new configuration with larger shelves (NEW).

The North American ice sheet seems to be more sensitive to the perturbation in the large shelves case (**Figure R6**a). However, the latter is mainly driven by climatic effects (**Figure R7**c). Although we could expect some inland ice dynamics change related to larger ice shelves due to a change in buttressing, this is not happening here since there is no drastic change in elevation that are due to ice divergence anomalies over the North American ice sheet (**Figure R7**f) with respect to our reference experiment (Figure 3 from the main text). The differences with the previous setup are mainly restricted to coastal areas.

[Figure]

Figure R7 : Same as Figure 3 from the main text but for the configuration with larger shelves.

The rationale behind the basal melt scaling approach is not very clear to me. If the intention is to investigate the response to subsurface warming, from a physical point of view it would make more sense to apply different temperature offsets instead and then derive basal melt from eq. (1). I guess that results would be quite different if a uniform temperature increase would be applied instead of the melt scaling.

The main reason why the melt scaling approach is not very appropriate is because where basal melt is very small in the control run, due to e.g. low water temperatures, even a scaling by a factor 100 or more will not necessarily result in a substantial increase of melt. In other words, a given scaling factor will result in a very different change in basal melt for different water conditions. This probably explains much of the differences seen in the response of the different ice shelfs to the perturbation experiments presented in the paper. This is acknowledged in section 4.1, but it remains unclear why the authors opted for the scaling approach in the first place.

Thank you for all of your comments. We will add the following explanation in the revised Perturbation experiments section :

*"The perturbation experiments were designed to be spatially consistent with the physics of the water masses. For instance, our design allows us to account for the fact that an abrupt change in AMOC likely leads to temperature changes in the AMOC's main areas of influence, rather than a uniform temperature change over the North Atlantic and Nordic Seas."*

Also, we wanted to be sure that it was possible for the system to amplify/dampen the perturbation away/toward the control state, to evaluate the sign of the feedback. To this end, we have chosen not to impose constant values either of basal melt rates nor oceanic temperatures.

A limitation of doing so is that the experiment results are dependent on the initial representation of the temperature in the ocean and model biases.

We agree that applying a constant temperature offset might overcome the model biases and is as such an interesting experiment to run. For this reason, we performed additional experiments, applying T'=T+$T_{offset}$ with $T_{offset}$=2,6,8,10 °C (**Figure R8**).

[Figure]

Figure R8 :  Ice sheet volume anomaly [$10^6$ km³] for (a) the North American, (b) the Greenland-Iceland and (c) the Eurasian ice sheets. Time series of (d) AMOC [Sv], (e) Freshwater input to the ocean model [Sv] and (f) NGRIP temperatures [°C] in perturbation experiments with $T_{offset}$=2,6,8,10 °C as well as the experiment with X=100.

These experiments are interesting as this way we force the North American ice sheet volume to decrease (yet not only from the Baffin Bay but also to its northern part). Discrepancies with the previous experimental setup also includes overshoots of the AMOC and NGRIP temperatures when the perturbation is halted. In the extreme case of +10 °C, the AMOC eventually shuts down, with no recovery after the perturbation ends. In this case, surface cooling and freshening is so large that convection stops in the North Atlantic Ocean yet develops in the North Pacific Ocean (**Figure R9**).

[Figure]

Figure R9 : Same as Figure 6 from the main text but for the experiment with $T_{offset}$=10 °C. Ice elevation is also depicted on (a),(d) for the control simulation and on (b) and (c), between 10-40 yr and between 470-500 yr after the perturbation starts. Ice elevation anomalies with respect to the control are depicted on (e) and (f).

These experiments with constant temperature offsets will be added to the revised article by replacing the experiments with constant melt rates (Figure 9 from the main text), as these are easier to connect to the real world. We will also add the following line to the Perturbation experiments section :

*"A limitation of the basal melt scaling approach is that the experiment results are dependent on the initial representation of the temperature in the ocean and model biases. To overcome this limitation, we perform additional experiments by applying a constant temperature offset within the formulation of the basal melt rate (section 4.1)."*

Some more details on the freshwater balance of the climate-ice sheet system are needed. Some additional information is provided in the SI, but I don't quite understand how the freshwater flux from the ice sheet can be ignored without resulting in a drift in ocean salinity. Water that is evaporating from the ocean will fall as precipitation over the ice sheets and, if ice sheets are in steady state, the same amount of water that is precipitating over the ice sheets has to reach the ocean as surface runoff, basal melt or calving in order to conserve water. If these freshwater fluxes to the ocean are ignored, this will lead to a drift in ocean volume and salinity. How is this prevented

in the noFWF experiments, including the spinup?
We acknowledge that the latter is not very clear in the main text and thank you for pointing this out.

It is to be noted that iLOVECLIM, the climate model, has its own hydrological cycle. This implies that surface runoffs are decomposed in two components : surface runoffs from iLOVECLIM and surface runoffs from GRISLI.

When there is no ice sheet coupling, or when the ice sheets are maintained constant, with V the ice sheet volume, then dV/dt = 0. Nevertheless, precipitation over land is still routed toward the ocean, such as an exchange between the atmosphere model and the ocean model to maintain a closed water budget in the climate model.

When the ice sheet coupling is activated, the iLOVECLIM hydrological cycle remains activated, ensuring that precipitations that fall over the ice sheet are routed toward the ocean. However, this time V can vary so we can use dV/dt as an additional freshwater flux. When dV is negative then freshwater is routed toward the ocean, such as an exchange between the ice sheet and the ocean model. dV is the sum of basal melt, calving and surface melt/accumulation. In the noFWF experiments, we ignore the transfer of dV (as a freshwater flux) from the ice sheets to the ocean but the iLOVECLIM hydrological cycle is still working.

This explanation will be added to the Supplementary Material.

Lines 228 to 230 will be rephrased as :

*"We perform another set of simulations, maintaining the oceanic perturbation and cutting off freshwater fluxes from the ice sheet model to the ocean model while the hydrological cycle of the climate model remains activated and precipitation that falls over land is still routed toward the ocean. In other words, here the ocean does not respond to the continental ice melt water and the associated oceanic feedback on the ice sheets is suppressed."*

**Minor comments**

L. 2: Please clarify that the forcing that is applied is actually not 'subsurface warming' but scaling of basal melt, which can produce very different results.
Thanks for your comment, this will be changed so there is no misconfusion. **The sentence will be replaced by the following : "*Using the iLOVECLIM climate model of intermediate complexity coupled with the GRISLI ice sheet model, we explore the consequences of an amplification of the melt rates beneath ice shelves on ice sheet dynamics and the associated feedbacks.*"**

L. 5: 'destabilisation' of what?
Thanks, we will remove this remnant of a previous version.

L. 26-27: should be kyr BP and not years BP.
Thank you, we will change it.

L. 53: again, this can be misleading and doesn't reflect what is done in the experiments.
Thanks, we will modify the sentence **for the following : "*In this work, we impose amplified oceanic melt rates to the 40 kyr BP Northern Hemisphere ice sheets to get insights on both past climate variability and mechanisms at play at the ocean-ice sheet interface.*"**

L. 72-73: Does this imply that ice shelves can not grow back after their thickness falls below 250 m?
It does not, the ice shelves can still expand as a result of upstream flow and surface accumulation. Thank you for pointing this out, we will add the following sentence to make this clearer :

*"It is to be noted, that after calving, the ice shelves are allowed to grow back, as a result of upstream ice flow and surface accumulation."*

L. 82: Does the model resolve the diurnal cycle?
No, the four hour time step in the atmospheric model is related to numerical stability.

**We will add the following sentence to the main text to comment on that : "*This time step allows for numerical model stability, as the ECBilt model does not resolve the diurnal cycle.*"**

L. 86-87: And what about the latent heat associated with basal melt? Is that not accounted for?
The latent heat associated with the basal melt is not taken into account in the model. It could be included in a future version. Still, according to the results of this study, we suppose that including it would affect the results by leading to a larger negative freshwater feedback on the continental ice volume losses.

Thank you, we will add this consideration to the revised version by adding the following sentence :

*"The latent heat flux associated with sub-shelf melt is not taken into account in the model, yet including it would lead to a larger cooling of the upper water column in this case."*

L. 93: Is the depth dependence of the freezing temperature not considered? As mentioned also in the introduction, this dependence can be important.
Indeed, it is not considered, the Millero formulation is a function of the local salinity and the depth z, but in the model z was set to a constant that is equal to 200 meters depth for every grid point in the freezing temperature formulation.

We have tested to what extent the depth dependance could change the results by taking the real model depth for each grid point rather than the 200 m constant. It appears that the results are not modified in this case (gray curves on **Figure R10**).

Yet, following your comment, we will keep the z dependance for the freezing temperature in future versions of the coupling. Thank you for pointing this out.

[Figure]

Figure R10 : Ice volume evolution for (a) the North American, (b) the Greenland-Iceland and (c) the Eurasian ice sheets. Times series of (d) AMOC in the North Atlantic, (e) Northern Hemisphere surface temperatures and (f) NGRIP surface temperatures for two experiments : X100$_{without}$ (ie. without depth dependance of the freezing temperature, same as in the main text) and X100$_{with}$ (ie. with depth dependance of the freezing temperature).

**We will modify this sentence in the main text to clarify this point : "*Where the local freezing temperature Tf =Tf (S, z = 200 m) is a function of salinity, following the formulation of Millero (1978). Here, the depth z is set to a constant value of 200 m, representing a characteristic shelf draft, to calculate the oceanic basal melt rate.*"**

L. 96: What is the target of the calibration procedure?
Thank you for the question. We have tested different values for the parameter $\gamma_T$ during the calibration phase (as well as other parameters related to the coupling). In the fully coupled model, in a large number of cases, ice sheets disintegrate (towards the pre-industrial ice sheets) or expand (towards the LGM ice sheets and beyond).

Consequently, at this stage of the model development, here we want to maintain the volume of the ice sheets at a level between these extreme states (PI or LGM-like ice sheets), this is the target. This may seem to be a wide range of possibilities for a target, but in the end, it has been quite difficult to achieve.

**We will add the following sentence to the main text to comment on that : "*This value for $\gamma_T$ ensures that the ice sheet volume remains intermediate between that of the Pre-Industrial (PI) period and the LGM, preventing excessive expansion or disintegration.*"**

L. 101-102: How is that done in the free-surface ocean model?
The CLIO model is said to be free-surface as the sea-surface elevation is solved. The reference level for the sea-surface elevation is chosen such that the average of the surface elevation anomalies over the globe is zero. Thus, the total volume of the ocean does not vary over time. The procedure is well described in Goosse , et al. (2001)., "Description of the CLIO model version 3.0.", sections 2.4 to 2.5.

L. 106: 'relatively stable conditions regarding June insolation at 65°N.'. What does that mean in the context of constant 40 ka orbital configuration?
Thank you for this question. We will clarify this sentence by adding the following paragraph :

*"We conduct a long coupled ice sheet-climate simulation under constant external forcing to derive the initial state. This approach assumes climate equilibrium, which is not realistic. However, the relatively small variations in greenhouse gas concentrations (GHG) and insolation at 65 °N around 40 kyr BP support the use of an equilibrium simulation for this specific time slice."*

See **Figure R11** below for the forcings.

[Figure]

Figure R11 : Insolation and orbital configuration forcings from Berger et al., (1978).

ky, k.y.,  k.y. B.P., ky B.P. are used interchangeably, causing some confusion. Please consistently use the same throughout the paper.
Thanks for this remark, we will change that.

L. 107-108: What does 'The ice sheet model is forced with sea level reconstruction' practically mean? Does sea level affect the bathymetry and land-sea mask? But the bathymetry in the climate model is set to LGM, so how are the two approaches combined?
The sea-level reconstructions are used by the ice sheet model to determine which areas of land are above water (ie. where the continental ice can possibly grow). These are not used by the iLOVECLIM model so it does not impact the bathymetry and land-sea mask.

We will add the following sentence to clarify this point :

*"The latter is only used by GRISLI to determine which areas of land are located above the sea level (i.e. where the continental ice can grow)."*

L. 107-108: Since constant values are used for GHGs and sea level, it would be clearer to state those values in the text.
You are right, thank you. We will add the values in the main text. The value for the GHG is set to 200 ppm and for the one for the sea level to minus 64 m. Thank you.

**The new sentence will be : " *Forcings are held constant at their 40 kyr BP values (namely 0.013°, 23.61° and 0.004 for eccentricity, obliquity and precession index, 220 ppm for the GHG concentration). The ice sheet model is forced with sea level reconstruction (-64 m at 40 kyr BP; Waelbroeck et al., 2002)."***

L. 125-127: This needs some clarification. Does that mean that during the spin-up phase the freshwater fluxes (runoff, basal melt and calving) from the ice sheet are ignored? I understand that the total ice volume change is small at equilibrium, but that doesn't imply that the freshwater fluxes are zero, just that total ice accumulation and ablation balance each other. I would think that it is important that the ocean 'sees' the freshwater fluxes from the ice sheets in the equilibrium initial model state.

Thank you for this comment. During the spin-up phase the freshwater fluxes from the ice sheet model to the ocean are indeed ignored. iLOVECLIM freshwater cycle is style activated, as described in a previous answer.

You are right that the freshwater fluxes (from the ice sheets to the ocean) are non zero at equilibrium (locally and over the total ice sheet), yet we have ignored them for the spin-up phase, the latter reason is detailed below.

The long spin-up is an accelerated experiment. We have chosen to do it this way as we initialize the spin-up with an LGM ice sheet and climate. It would have taken a long (real) time to equilibrate the fully coupled model to 40 kyr BP conditions without the acceleration since ice sheets have a slow dynamic, in comparison with ocean and atmosphere. Indeed, when the full coupling is activated and with no acceleration, we are able to run ~400 years per day.

However, when using an acceleration factor (here of 10 years), we cannot conserve both total ice mass change and the rate of mass change, as explained in Supplementary S2. If we conserve the total mass change, then the ocean model would receive the equivalent of 10 consecutive years of melting (and the freshwater fluxes would be 10 times larger than the annual mean flux). This could lead to AMOC shutdown, especially in the beginning of the spin-up simulation as we start from an LGM ice sheet. Conversely, if we provide CLIO with the mean freshwater flux, we only account for one-tenth of the total mass change and it could still be a large flux (again as we start from an LGM ice sheet).

The best we could do in a future version, could be either to perform the spin-up feeding CLIO with an annual mean flux from GRISLI or extend the spin-up simulation with synchronous coupling and freshwater exchange between the ice sheet and the ocean for several hundreds of years.

Nevertheless, here, when we extend the spin-up simulation with synchronous coupling and including the freshwater fluxes from the ice sheets to the ocean, the drift is quite small (see the **Figure R12** below or for instance the mean difference in global mean salinity between CTRLfwf and CTRLnofwf on Figure S2 from the Supplementary Material).

[Figure]

Figure R12 : Ice volume evolution for (a) the North American, (b) the Greenland-Iceland and (c) the Eurasian ice sheets. Times series of (d) AMOC in the North Atlantic, (e) Northern Hemisphere surface temperatures for two simulations. In **blue** : the long spin-up simulation (asynchronous, so 8,000 years for the climate and 80,000 for the ice sheets, with no freshwater fluxes from the ice sheet to the ocean) ; in **orange** : an extended simulation of 2,000 years that is synchronous and includes freshwater fluxes from the ice sheet to the ocean.

Comparing the last thousand years of the spin-up and for the first hundred years of the extended run that includes freshwater fluxes from the ice sheet to the ocean, the North American ice sheet's volume goes from 18.14 to 18.06 millions of km³ (-0.5%) ; the Greenland-Iceland volume goes from 4.259 to 4.258 millions of km³ (-0.02%) ; the Eurasian volume from 1.006 to 1.007 millions of km³ (+0.06%) ; AMOC reduction is of 5%, from 22.15 Sv to 21.09 Sv and Northern Hemisphere temperature decrease is of 0.2%, from 12.45 °C to 12.39 °C.

Thank you for pointing this out. This will be further detailed in the Supplementary Material.

We will also add the following sentence to the description of the spin-up simulation.

*"During the spin-up phase the freshwater fluxes from the ice sheet model to the ocean are ignored. This approach is necessary because we cannot conserve both total ice mass change and the rate of mass change when using an acceleration factor (Supplementary S3). However, at the end of the spin-up phase the ice sheet is in equilibrium with the rest of the climate system. Therefore the ice sheet volume is quasi stable and produces a net zero freshwater flux to the ocean. There is thus no significant discontinuity in our control experiment with ice sheet freshwater flux included when branched from the spin-up simulation."*

L. 141-142: Not fully clear what is meant here. I don't think that a new equilibrium can be reached in only 500 years.
We will modify this sentence by *"before reaching a new state with minimal ice volume change for the Greenland-Iceland ice sheet"*, thank you for noticing it.

L. 177-178: Would be interesting to see the separate contributions of basal melt and calving. This should also be discussed in some more detail in relation with the calving law that is applied in the ice sheet model, which I guess could result in thick shelf ice being cut instantaneously if basal melt lowers the ice thickness below 250 m. How realistic is that?

L. 178: The equivalent number in Sv units would be useful, as that is more commonly used when it comes to freshwater forcing of the ocean. Moreover, this number should depend on the magnitude of the basal melt scaling factor, no?
We will change the number to Sv units, thank you. You are right that this number depends on the intensity of the scaling factor, the numbers were given for the experiment with X=100. The sentence will be corrected.

250 m is a common magnitude order for ice shelves thickness at the terminus.

**We will add the following sentences to the Supplementary Material section about the calving law to comment on that : "*In the ice sheet model, the calving law is simple and is based on a minimum cutoff threshold. It is easy to implement and it ensures that the ice shelves do not become thinner than a minimal value (hereafter referred to as hcutpla). Indeed, thin ice shelves are more prone to breakage due to processes that are not represented in the model, such as ice shelf rifting. The cutting threshold for the ice shelf thickness above the plateau, hcutpla, is set to 250 m, which reflects a typical value of the observed ice shelf thickness near the terminus in Antarctica today.*"**

L. 204-206: This sounds very speculative. Is it not more likely that the warming is a result of the transition from perennial ice shelf to seasonal sea ice cover?
Following your comment we have checked this a little bit further, thank you for noticing it.

More freshwater is added locally in the perturbed experiment than in the ctrl run at the end of the simulations (**Figure R13**), so our suggestion was wrong.

[Figure]

Figure R13 : Freshwater input [Sv] to the ocean model and annual sea-ice contour corresponding to 5,20,40,60 and 80% averaged over 470-500 years for (a) the control experiment and (b) the perturbation experiment with factor X=100.

We see that along the perturbation period, the yearly sea ice cover remains 0 after ~100 years. We also see that the temperature and salinity differences with the control occur over the whole water column at the chosen location (not only near the surface ; **Figure R14**).

[Figure]

Figure R14 : Hovmoller diagrams for the profile whose location is the blue dot in Figure R13b. For the perturbation experiment with factor X=100 (resp. the control experiment), (a) (resp. (c)) absolute salinity [g/kg] and (b) (resp. (d)) conservative temperature [°C].

This led us to realize that the positive salinity and temperature anomalies, described lines 204-206, actually result from a resumption of the AMOC and renewed influx of heat and salt of Atlantic origin following the period of maximum AMOC slowdown. As we can see on the **Figure R15** below (that is the same as Figure 7 of the main text but with salinity anomalies included), the area of positive salinity anomaly at the end of the experiment is located not only at the surface but also at the subsurface along the Atlantic Water pathway.

[Figure]

Figure R15 : Conservative temperature [°C] at 300 m depth horizon, (a) averaged over the control experiment, and associated anomaly between the perturbation experiment with X=100 (b) over 10-40 yrs and (c) over 470-500 yrs and the control. (d,e,f) Same for the absolute salinity [g/kg].

This sentence will be replaced by the following :

"*The positive anomalies therefore suggest a resumption of the AMOC and renewed influx of heat and salt of Atlantic origin following the period of maximum AMOC slowdown.*"

L. 260-262: Do you really account for the difference in temperature of the meltwater relative to the ocean water temperature? Or is the described cooling due to the latent heat extracted from the ocean to melt the ice?
No we don't, freshwater is added at the temperature of the surrounding waters. We agree that this was not clear in the manuscript. In this sentence the cooling was referring to the cooling in the ocean following the freshwater addition (Figure 6e,7c).

Thank you for noticing it, the sentence will be replaced by the following :

"*Meltwater fluxes into the ocean bring water that is fresher than the surrounding ocean and favor sea-ice expansion, increasing stratification and reducing vertical mixing.*"

L. 334: Why 'contradiction'?
We used 'contradiction' in the sense that our Laurentide ice sheet doesn't display any significant volume variations although paleo-data have shown that it was the case for this time period. We will replace the sentence by : "*There are several possible reasons for this discrepancy between the simulated behavior and the observations :*"

Fig. 6: I guess that 5 m depth is equivalent to the surface? Or how thick is the top ocean model layer? Panels g-i seem to show depth of convection rather than density, as stated in the caption.
Yes it is, the first layer is centered at 5 m depth and is 10 m thick.
Panels g-i are indeed the depth of convection, sorry for the mistake in the caption, we will modify it.

**Specific comments**
L. 48: increases -> increase
L. 284: others -> other
Thanks, these mistakes will be corrected.

Thank you for all the comments and questions,

Louise Abot, on behalf of all co-authors.
**References**

Alvarez-Solas, Jorge, et al. "Links between ocean temperature and iceberg discharge during Heinrich events." *Nature Geoscience* 3.2 (2010): 122-126.

Alvarez-Solas, Jorge, et al. "Iceberg discharges of the last glacial period driven by oceanic circulation changes." *Proceedings of the National Academy of Sciences* 110.41 (2013): 16350-16354.

Barker, Stephen, et al. "Icebergs not the trigger for North Atlantic cold events." *Nature* 520.7547 (2015): 333-336.

Bond, Gerard, et al. "Evidence for massive discharges of icebergs into the North Atlantic ocean during the last glacial period." *Nature* 360.6401 (1992): 245-249.

Bond, Gerard, et al. "Correlations between climate records from North Atlantic sediments and Greenland ice." *Nature* 365.6442 (1993): 143-147.

Bond, Gerard C., and Rusty Lotti. "Iceberg discharges into the North Atlantic on millennial time scales during the last glaciation." *science* 267.5200 (1995): 1005-1010.

Berger, André. "Long-term variations of caloric insolation resulting from the Earth's orbital elements." *Quaternary research* 9.2 (1978): 139-167.

Dansgaard, Willi, et al. "North Atlantic climatic oscillations revealed by deep Greenland ice cores." *Climate processes and climate sensitivity* 29 (1984): 288-298.

Dansgaard, Willi, et al. "Evidence for general instability of past climate from a 250-kyr ice-core record." *nature* 364.6434 (1993): 218-220.

Ditlevsen, Peter D., Katrine Krogh Andersen, and Anders Svensson. "The DO-climate events are probably noise induced: statistical investigation of the claimed 1470 years cycle." *Climate of the Past* 3.1 (2007): 129-134.

Elliot, Mary, et al. "Millennial-scale iceberg discharges in the Irminger Basin during the last glacial period: Relationship with the Heinrich events and environmental settings." *Paleoceanography* 13.5 (1998): 433-446.

Goosse, Hugues, et al. "Description of the Earth system model of intermediate complexity LOVECLIM version 1.2." *Geoscientific Model Development* 3.2 (2010): 603-633.

Gowan, Evan J., et al. "A new global ice sheet reconstruction for the past 80 000 years." *Nature communications* 12.1 (2021): 1199.

Heinrich, Hartmut. "Origin and consequences of cyclic ice rafting in the northeast Atlantic Ocean during the past 130,000 years." *Quaternary research* 29.2 (1988): 142-152.

Johnsen, S. J., et al. "Irregular glacial interstadials recorded in a new Greenland ice core." *Nature* 359.6393 (1992): 311-313.

Jonkers, Lukas, et al. "A reconstruction of sea surface warming in the northern North Atlantic during MIS 3 ice-rafting events." *Quaternary Science Reviews* 29.15-16 (2010): 1791-1800.

Kindler, Philippe, et al. "Temperature reconstruction from 10 to 120 kyr b2k from the NGRIP ice core." *Climate of the Past* 10.2 (2014): 887-902.

Max, Lars, et al. "Subsurface ocean warming preceded Heinrich Events." *Nature Communications* 13.1 (2022): 4217.

Mignot, Juliette, Andrey Ganopolski, and Anders Levermann. "Atlantic subsurface temperatures: Response to a shutdown of the overturning circulation and consequences for its recovery." *Journal of Climate* 20.19 (2007): 4884-4898.

Rasmussen, Tine L., and Erik Thomsen. "The role of the North Atlantic Drift in the millennial timescale glacial climate fluctuations." *Palaeogeography, Palaeoclimatology, Palaeoecology* 210.1 (2004): 101-116.

Schulz, Michael. "On the 1470-year pacing of Dansgaard-Oeschger warm events." *Paleoceanography* 17.2 (2002): 4-1.

**Detailed response to Reviewer #2's comment**

The study uses coupled-climate ice sheet modelling to assess the interactions between ocean subsurface warming and ice sheet dynamics during the last glacial period (Northern Hemisphere ice), with a particular focus on basal melt rates and freshwater release. The authors describe the impact of the basal melt of coastal ice on ice sheet geometry, and explain how the response of regional sea-ice to freshwater reduces local oceanic convection, with consequences for AMOC. The overall effect of the freshwater is a negative feedback to ice mass loss in Greenland and Eurasia, although the authors find the North American ice sheet to be stable anyway.

The language used is clear to follow as, for the most part, are the figures. I enjoyed reading it. However, there are some parts of the framing that need tightening up before publication, to properly situate the work, making sure its relevance/importance can be appropriately contextualized by the reader and thus built on, by future work in the community. There is a bit too much assumed contextual knowledge/perspective. I think with some improvements in the presentation of the work (as suggested more fully below), the work will add some value to the field.

Thank you for the time you spent reading the manuscript and your detailed analysis.

In the whole framing of the study, the relation between the experiment design (and results) and the past glacial cycle is somewhat vague; it is not very clear specifically what, beyond a thought experiment for understanding this coupled-models behavior, is being tested. The final paragraph on page 2 explains the specific aims, but this needs relating back to the real world. Discussion section 4.3 has some relation to real past events, but it is short, and vague, and comes only at this late stage in the manuscript.

Thank you for this comment. We will clarify the sequence of ideas, the scientific questions we are addressing, as well as their connections with the real world, in the following response as well as in the main text's introduction and description of the experiments.

The motivation of this study is understanding the cause of the massive iceberg discharges that occurred during the Last Glacial Period. As it was not very clear, we will add the two following paragraphs at the beginning of the introduction :

*"The Last Glacial Period began around 75 kyr before present (BP) and ended with the Last Glacial Maximum (LGM) around 21 kyr BP. An intriguing feature of the last glacial period is its millennial-scale variability, first revealed in Greenland ice cores, known as the Dansgaard-Oeschger (DO) cycles (Dansgaard et al., 1984, 1993). These are transitions between relatively cold (stadial) and warm (interstadial) periods in the Northern Hemisphere (Johnsen et al., 1992). The rapid warming (or DO event) generally takes place over a period of several decades and is of the order of 10oC at the North Greenland Ice Core Project (NGRIP) site (Kindler et al., 2014). These events occurred at a frequency of around 1,500 yr (Schulz, 2002), although this frequency is debated (Ditlevsen et al., 2007). They are followed by a slower cooling of the order of a hundred to a thousand years, and the cycle ends with a sudden cooling (Kindler et al., 2014).*

*Some of the stadials are accompanied by massive iceberg discharges from the Northern Hemisphere ice sheets in the North Atlantic Ocean, at a frequency of several thousand years (Heinrich, 1988; Bond and Lotti, 1995). The iceberg discharges are identified in marine sediment cores by one or several layers of ice rafted debris, the materials entrapped and transported by drifting ice (Bond et al., 1992; Bond and Lotti, 1995; Elliot et al., 1998). The iceberg discharges are referred to as Heinrich Events (HE) when the continental ice is originating from the Laurentide ice sheet, the eastern part of the North American ice sheet. The Heinrich events mostly occur after several DO cycles and are followed by a DO event (Bond et al., 1993). Barker et al. (2015), through the examination of a site southwest of Iceland, suggested that massive iceberg discharges were a consequence of prolonged stadial conditions. It was suggested that massive iceberg discharges occur after an initial reduction of the Atlantic Meridional Overturning Circulation (AMOC) and the build up of a heat reservoir at the subsurface in the North Atlantic from both observations and modelling studies (e.g., Jonkers et al., 2010; Max et al., 2022; Mignot et al., 2007)."*

To date, there is no scientific consensus on the ultimate cause behind massive iceberg discharges. For instance, some studies suggest self-sustained ice sheet oscillations, while others propose a perturbation triggered by changes in ocean conditions. The focus of our study is to test the latter hypothesis, using a coupled ice-climate model, namely iLOVECLIM-GRISLI. We will add the three following sentences in the introduction to clarify this :

*"Nonetheless, there is no scientific consensus on the ultimate cause behind the massive iceberg discharges to date. For instance, some studies have suggested that self-sustained ice sheet oscillations are at play (MacAyeal, 1993), while others propose that the discharges were triggered by an external forcing such as a change in ocean conditions (Alvarez-Solas et al., 2010). The focus of our study is to test the latter hypothesis, using a coupled climate-ice sheet model, namely iLOVECLIM-GRISLI."*

As explained in the introduction of the article (lines 26 to 42), it was suggested that a rise in subsurface ocean temperature during a stadial could have led to the ice discharges. However, previous modelling studies focusing on ice-ocean interactions in the context of massive iceberg discharges have relied either on an ice sheet model with oceanic forcing or on a climate model forced by a freshwater flux (hosing experiments). The novelty of our study lies in the use of a coupled model.

Previous studies using iLOVECLIM alone (Friedriech et al,2010) or GRISLI alone (Álvarez-Solas, 2011) have shown abrupt variations in ocean circulation or ice sheet discharges. However, in our setup, the iLOVECLIM-GRISLI coupling does not exhibit abrupt variations, neither in the ice sheets nor in the ocean circulation, under 40 kyr BP constant or transient forcings. This is why we conducted perturbation experiments at the interface. With the fully-coupled model, we can examine whether a perturbation at the interface is being amplified or dampened. We will add the following paragraph in the introduction to explain why we designed perturbation experiments.

*"In our setup, the iLOVECLIM-GRISLI coupled model, under constant or transient forcings corresponding to the Last Glacial Period, does not exhibit abrupt internal oscillations, neither in the ice sheets nor in the ocean circulation. Therefore, in this study we chose to conduct perturbation experiments at the ice-ocean interface. With the fully-coupled model, we can examine whether a perturbation at the interface is being amplified or dampened."*

In this vein, I find the final sentence in the abstract hard to understand: what is it specifically that the presented results show, and what here is new, or how precisely does the study feed into an existing body of evidence? What is meant by 'more accurately constrained mode results'? Constrained how and by what? And how/what would more accurate model constraints help with? Which 'past changes' or 'future ice sheet behavior' specifically are being referenced here? I'm not questioning the value of the study, but the framing needs clearer precision so that I understand precisely what message(s) to take away. That ice-ocean interactions are complex are reasonably well established with a very wide body of precise literature already.

The main point to take away is that, at the ocean-ice sheet interface, the feedback of freshwater release on ice sheet discharge is overall negative. This feedback sign was not obvious when we designed the experiments; in fact, we initially expected it to be positive. This would have implied that the ocean, through the activation of a positive feedback loop following an initial perturbation, would warm at the shelf drafts, leading to increased ice discharge –  a mechanism that might have led to massive iceberg discharges of the Last Glacial Period.

As explained above, this study is motivated by our will to understand how the massive iceberg discharges during the Last Glacial Period could have been triggered by oceanic processes. More broadly, the study aims to explore interactions between ice sheets and ocean, regardless of the time period, such as a process study. In this context, the last phrase of the abstract (*"to enhance our understanding of past changes and the predictions of future ice sheet behavior and sea level rise"*) reflects the traditional goal of "understanding the past to better anticipate the future". However, we agree that this phrasing is too vague, we have removed this portion of the sentence to clarify the paper's focus and primary motivation.

By *"more accurately constrained model results"*, we were referring specifically to the last sentence of the conclusion (*"Lastly, our model can simulate ocean carbon isotopes (Bouttes et al., 2015), allowing for direct comparison with observational data from marine sediment cores and providing additional constraints."*). There were no specific constraints on the ocean to calibrate the coupled model. So, another idea for future model development is to focus on ocean conditions and calibrate the ocean with carbon isotopes.

To clarify this, we will replace the final sentence of the abstract by : *"This study highlights that the ocean-ice sheet interactions following an episode of subsurface warming may be characterized by negative feedbacks, thus dampening iceberg discharges. This emphasizes the need for further research using fully coupled*

*models to explore the triggering mechanisms of massive iceberg discharges and to clarify the role of the ocean in these events."*

**Specific points:**

**Introduction:**

Is too short. It lacks detailed information on what is known about the processes of ice-ocean interactions, and at different scales, or what debates/known-unknowns/limitations exist within that topic. What is included is at a very large macro-level, but what are the important dynamics in grounding line-processes, and what about ice sheet cavity-scale processes in the ocean? Thank you for this comment. We will add more information on the marine ice sheet instability to the introduction section describing ice-ocean interactions (lines 43 to 49) :

*"Another positive feedback is the marine ice sheet instability, which is linked to the dynamics of the grounding line. The mathematical formulation of the ice flux at the grounding line suggests that this flux increases with the ice thickness (Schoof, 2007). When a marine-terminating ice sheet sits on an upward-sloping bed toward the ocean, an initial retreat of the grounding line causes ice thickening and acceleration of the ice flow at the grounding line, which in turn drives further retreat. This process may have played a role in Heinrich events given the bedrock's shape at the Hudson strait mouth (Schoof, 2007)."*

Feedback mechanisms arising from ice sheet-cavity scale processes are not represented in our model. In fact global cavity-enabled ocean models that include an interactive ice sheet model are currently being developed in the community but their applications are still limited to present-day climate (e.g. Mathiot et al., 2017). Cavities clearly play a role in the global ocean, but it is difficult at present to have a good picture of their related feedbacks. Nonetheless, following your advice, we have added the following in the introduction :

*"Additional feedback mechanisms could emerge from ice sheet-cavity scale processes, such as tidal forcing or buoyancy-driven circulation beneath ice shelves (e.g., Makinson et al., 2011; Gwyther et al., 2016). For example, a rise in ocean temperature at the base of the ice shelves could enhance the meltwater production, thereby strengthening the circulation within a cavity, leading to an increased heat supply and further melting (Gwyther et al., 2016). However, uncertainties related to these processes remain large since very few cavity-enabled ocean models can operate at the global scale. For this reason and given our coarse resolution ocean model, we do not discuss the cavity recirculation feedbacks any further in this paper."*

**Methods:**

Is GRISLI an appropriate tool for addressing marine ice sheet interactions? From my background knowledge, I believe it is, but this is not clear from the manuscript. Further justification/evidence of this is needed.

Yes, GRISLI is an appropriate tool, we will add the following sentence to the Model and Methods : *"GRISLI is an appropriate tool to address ice sheet dynamics and interactions with the oceanic component as it explicitly calculates ice stream velocities, the position of the grounding line and the the behavior of ice shelves. For instance, GRISLI has shown a good ability to reproduce grounding line migrations and ice volume changes in Antarctica during the last 400 kyr BP (Quiquet et al., 2018,Crotti et al.,2022)."*

Experiment design needs more justification/explanation, e.g.:

- Initial Condition section:

(a) Why 40 ka BP? Not enough to say it is relatively stable and precedes HS4 when no background is given on why this is important. There are other stable periods. What is HS4? Is LGM bathymetry appropriate? What is meant by 'sea level forcing'? Thank you for these questions. Heinrich stadials are now defined in the Introduction section (see above) and we will add the following sentences to the Initial condition section :

*"We conduct a long coupled ice sheet-climate simulation under constant external forcing to derive the initial state. This approach assumes climate equilibrium, which is not realistic. However, the relatively small variations in greenhouse gas concentrations (GHG) and insolation at 65 °N around 40 kyr BP support the use of an equilibrium simulation for this specific time slice. Furthermore, this period aligns with Heinrich Stadial 4 (HS4), which extends from 39.85 to 38.17 kyr B.P. (Waelbroeck et al., 2019). "*

See **Figure R1** below for the evolution of the forcings. Thus, our 1000 yr long simulations following 500 yr long perturbations applied at 40 kyr BP may be compared to HS4 observations.

We will also add the following in the same section about the sea level forcing :

*"The ice sheet model is forced with sea level reconstruction (-64 m at 40 kyr BP; Waelbroeck et al., 2002). The latter is only used in the GRISLI model to determine which areas of land are located above the sea level (i.e. where the continental ice can grow)."*

As well as this consideration about the LGM bathymetry :

*"However, since sea level and bathymetry at 40 kyr BP have intermediate value between the LGM and the Pre-Industrial (PI) values, we also test the sensitivity of the results to the Pre-Industrial bathymetry in the discussion section. Thus, two extreme bathymetry possibilities are investigated."*

[Figure]

Figure R1 : GHG concentration (Lüthi et al., 2008), as well as insolation and orbital configuration forcings (Berger et al., 1978).

(b) Figure 1: shades of gray are hard to see
Thank you, we removed the shades of gray (**Figure R2**). We will also add shelf drafts and basal melt rates, following a comment of reviewer #1, so the new Figure 1 will be :

[Figure]

Figure R2 : 40 kyr equilibrium. Ice sheet elevation (shades of blue) [m] at the end of the equilibrium simulation. Light blue contour is the mean position of the grounding line at the same time. Yellow contours are the minimal and maximal ice sheets

extent in the reconstruction from Gowan et al. (2021), these differ only over the central North American ice sheet. Areas used to derive regional ice volumes on Figure 2 are shaded in red, green and magenta for the North American, Greenland-Iceland and Eurasian ice sheets respectively. The black lines represent topographic contours at intervals of 500 meters. (b) Ice shelf drafts [m] and (c) basal melt rates beneath the shelves [m/yr] at the end of the equilibrium simulation.

(c) Line 117: 'corresponds rather well to the extent of the reconstructed ice sheet.' Not in my opinion, not for North American or Eurasia. Greenland also looks too large. I see big differences, and your total ice volume is (at most) almost twice as large as the reconstructed estimate. A more thorough and transparent assessment of agreement is needed here, AND a thorough and transparent discussion of what the reconstruction actually is – how certain/uncertain is that (where, when and why). Thank you for this comment. We will replace the paragraph of the Initial condition section by the following :

*"We obtain a total volume of 24 million km³ for the Northern Hemisphere ice sheets, which corresponds to 48 mSLE. In comparison, the reconstruction from Gowan et al. (2021) presents volumes of around 12 and 16 million km³ for minimal and maximal scenarios, corresponding to 28 and 37 mSLE respectively. However, Gowan et al. (2021) acknowledge that their total ice sheet reconstructions for the period between 57 and 29 kyr BP (MIS3) do not align with proxy-based sea level reconstructions. Specifically, they estimate a global sea level that is 20 m higher (maximum scenario) than the reconstructed value of approximately −60 to -90 m at 40 kyr BP (Waelbroeck et al., 2002; Arz et al., 2007; Siddall et al.,2008).*
*Also, the evolution of the Northern Hemisphere ice sheets remains uncertain prior to the Last Glacial Maximum. Reconstructions often rely on inverse methods that use observational data and estimates of global mean sea level, sometimes supported by numerical ice sheet modelling (e.g.; Marshall et al., 2000; Kleman et al., 2010; Gowan et al., 2021; Pico et al., 2018; Dalton et al., 2022b). Significant discrepancies exist between the different reconstructions. For example, the question of whether the Hudson Bay area was glaciated around 40 kyr BP and to what extent remains debated (e.g.; Batchelor et al., 2019; Miller and Andrews, 2019). Given these uncertainties, we conclude that our simulated ice sheet volume falls within an acceptable range for the 40 kyr BP time slice."*

- Perturbation experiment:

(a) why did you drop the acceleration and move to 1:1 run speed for both models? The acceleration is only used for the spin up simulation, in order to equilibrate ice sheets with the climate, as the ice sheets take a long time to adjust, which is too computationally expensive in real time. For example, the 8,000-year simulation would have run for around 20 days with the 1:1 speed.

**We will add the following sentence to the main text : "*The acceleration is only used for the spin-up simulation, in order to equilibrate the ice sheets with the climate, as the ice sheets take a long time to adjust.*"**

(b) What does the applied range in basal melt amplification factor correspond to in a non-technical (model) sense, i.e. in the real world? How was the range determined? How is it justified?

We agree that the experiment design should be better explained. Thank you for pointing this out. We will add the following paragraphs in the Perturbation experiment section :

"*This scaling (X) of the basal melt is equivalent to imposing a temperature anomaly, that depends on the water masses and background state of the ocean, at the shelf drafts. Specifically, we apply a perturbation whose intensity is proportional to the local temperature anomaly with respect to the freezing temperature, $T_f$.*

*Indeed, when $T > T_f$ , adding a perturbation term $\delta T$ to T, defined by $\delta T = \mathcal{E}(T-T_f)$ (with $\mathcal{E}>0$) leads to the following oceanic basal melt rate : $OBM=\gamma_T A(T+\mathcal{E}(T-T_f)-T_f)^2 = \alpha(1+\mathcal{E})^2(T-T_f)^2$. And we rewrite $X=(1+\mathcal{E})^2$. So there is a direct correspondence between X and the temperature anomaly intensity $\mathcal{E}$ seen by the ice shelves. For example, a doubling of the temperature anomaly to the freezing point corresponds to $\mathcal{E} = 1$ and $X = 4$. A 10 time increase of the temperature anomaly corresponds to $\mathcal{E} = 9$ and $X = 100$.*

*This way, the perturbation experiments aim to represent an increased heat flux to the ice shelf drafts, reflecting subsurface ocean warming in the North Atlantic during a Dansgaard-Oeschger stadial (e.g., Rasmussen et al., 2004). The perturbation experiments were designed to be spatially consistent with the physics of the water masses. For instance, our design allows us to account for the fact that an abrupt change in AMOC likely leads to temperature changes in the AMOC's main areas of influence, rather than a uniform temperature change over the North Atlantic and Nordic Seas.*"

The range for X was determined empirically when we applied this method to perturb the ice sheet dynamics.

(c) Are there implications for applying the subsurface warming perturbation as a basal melt amplification factor for the response of the ocean to the resultant freshwater flux? i.e. the thermal profile of the ocean does not have subsurface warming, initially, so what does this mean for the ocean structure/stability (in terms of buoyancy profile and perturbation), both in terms of the initial ocean condition (including convection) and the response of the ocean to ice sheet freshwater fluxes? This also needs revisiting in the results/discussion.

The ocean model simply adjusts its temperatures and salinity in response to the freshwater input, with associated impacts on the density profile, vertical convection and hence ocean circulation.

Initially the thermal profile of the ocean does not have subsurface warming, the ocean structure at year (before the perturbation is triggered) is the same in the control simulation and in the perturbation experiment, as both simulations are branched on the same spin-up simulation (**Figure R3**a,b).
When the perturbation is triggered, the ocean temperatures and salinities vary, which can modulate the perturbation in the vicinity of the ice shelves.

For instance, the Labrador Sea is a convective area. The initial scaling of the basal melt leads
to freshening and cooling of the near surface, while the subsurface (~300 m) warms. The vertical density gradient increases, so does the stratification. After some time, convection resumes (See for instance Figure S6 from the Supplementary Materials).

[Figure]

Figure R3 : Convection depth [m] (a) averaged along the CTRL simulation and (b) averaged over 0-5 years in the perturbation experiment with X = 100. (c,d,e) Profiles of density [kg/m³], ocean temperature [°C] and absolute salinity [g/kg] in the CTRL and in the perturbation experiment with factor X = 100 at different times after the perturbation is triggered. Blue shading corresponds to one standard deviation of the CTRL simulation. The location of the profiles is the red dot on (a,b). On (a,b) red contours are 100 and 300 m isocontours of mean convection depth, white and blue are the 30% sea ice extent for the CTRL and for the first 5 years of the perturbation experiment with X = 100.

We will add the three following italicized sentences to the last paragraph of the Perturbation experiments section (lines 130 to 132) in order to clarify these aspects :

From the ice sheets point of view, increasing the oceanic basal melt rates is equivalent to imposing a subsurface warming yet this has no effect on initial ocean temperatures. *"Initially, the ocean structure is the same for the control and perturbed experiments, as the simulations are branched on the same spin-up simulation."* Since we use a coupled model setup, the ice sheet retreat induced by the perturbation impacts the ocean through the resulting freshwater flux. *"The ocean model adjusts its temperatures and salinity in response to the freshwater input, with associated impacts on the density profile, vertical convection and hence ocean*

*circulation. Thus, the ocean model response can modulate the perturbation in the vicinity of the ice shelves.”*

**Results:**

- How well are the ice streams depicted in the model? This may also be a question about model ability, or resolution. The ice velocity maps show a wide splurge of acceleration in ice velocity, but I'm struggling to pick out many streams (with the few exceptions mentioned at the bottom of page 6). Please assess and comment (critically) on this in the revised manuscript.

Thank you for this comment. The velocities are smoothed over the topography in comparison with higher resolution models, yet GRISLI does a reasonably good job in depicting the main streams. The smoothed patch for velocities might come from the colormap we are using, see **Figure R4** below, with a different colormap and a linear colorbar.

At the LGM, the modeled ice sheet reproduces the major features of the ice streams of the North American ice sheet. Following the nomenclature from Margold et al., (2015) around the Baffin Bay the Hudson Strait ice stream at 60°N, some minor streams at the location of the current Baffin Island, the Lancaster Sound ice stream and the Nares Strait ice stream are depicted. To the north, the Kennedy-Robenson Channel at 50°W, the Nansen Sound (between 80-100°W) are also represented. Further west, there is a sluggysher patch corresponding to the Massey Sound and Prince Gustaf Adolf Sea ice streams. Between 120 and 140 °W, the M'Clure Strait and Amundsen Gulf ice streams are depicted (**Figure R4**a,b,c). To the south of the North American ice sheet, on the contrary, there are no well-identified ice streams. The representation of the ice streams is quite similar at 40 kyr BP in our model (**Figure R4**d,e,f).

We will add the following sentences to the main text :

*"Despite its low resolution (40 km), GRISLI is capable of representing large-scale features such as major ice streams at the LGM (e.g., Quiquet et al., 2021). The major ice streams, such as the Hudson Strait, the Lancaster Sound and Amundsen Gulf ice streams (nomenclature follows Margold et al., 2015) remain present at 40 kyr BP. To the south of the North American ice sheet, on the contrary, there are no well-identified ice streams.”*

[Figure]

Figure R4: Ice sheet elevation above sea-level [m] modeled (a) at the LGM and (b) at 40 kyr BP. Ice speed [m/yr] (b) at the LGM and (e) at 40 kyr BP. (c,f) are zooms of (b,d).

- page 8 (around line 175) – yes, but we come back to the question of what 'X >= 100' actually means in real-world terms. What does this equate to in terms of subsurface warming/ocean structure change to drive the ice sheet change? This needs relating back to real conditions, rather than simply technical model components, to offer genuine insight for understanding the ocean-ice interaction. Thank you for pointing this out. Please see our response above to your first comment on this subject. This sentence will be modified into :

*"Together with positive anomalies of the surface mass balance to the south, this contributes to the small thickening of the North American ice sheet for experiments with at least a 10 time increase of the temperature anomaly, with respect to the freezing temperature, at the shelf drafts (X ≥ 100 ; Figure 2)."*

- Line 181: ice sheet stabilizing mechanisms such as…? Please diagnose and include these, and then relate them to the real world.
Thank you for your comment, we will replace this sentence by the following ones :

This results in a freshwater flux that decreases with time (but remains significant). *"This decrease is partly attributed to the reduced ice volume exposed to basal melting as ice shelves thin or vanish in certain regions, such as Svalbard and Iceland. It also suggests the presence of ice sheet stabilizing mechanisms within the model, that is, a negative feedback that mitigates ice volume loss. Here, this feedback involves a decrease of the temperature at shelf drafts, which in turn reduces the basal melt rates."*

- Figure 5: sparks the question, what should the conditions be for these depicted metrics (based on the real, past world)? Needs showing/discussing as part of analysis.
Also, can differences in timings/temporal evolution of the different variables plotted

be highlighted and explained.

*Thank you for this question. We will comment on this in the revised manuscript :*

*"For most of these metrics, there are no past estimates available. Greenland temperatures at the NGRIP sites is the only exception (Kindler et al., 2014). At this location, the authors obtain around -40 to -45 °C around 40 kyr BP, while we simulate warmer temperatures around -35 to -40 °C. Paleoproxy records show that the AMOC was in a much lower regime during stadials than interstadials but do not provide quantitative estimates in Sv (Gottschalk et al., 2015; Henry et al., 2016, Waelbroeck et al., 2018)."*

*Following your comment, we went a bit further in the temporal evolution/timings analysis between the different variables.*

*Considering times series with 21 years centered running mean, the maximum changes for the sea ice extent (+$10^{12}$ km² in comparison with the control), the AMOC (−5 Sv), the NGRIP (−0.53 oC) and Northern Hemisphere (−0.49 °C) temperatures are reached after 14, 18, 21 and 33 years for the experiment with perturbation factor of 100, for instance. Differences in timings of the different variables are difficult to establish as the times series are noisy.*

*Considering the smoothing of 21 years and normalized time series, the maximum lagged correlation between the AMOC and sea-ice extent is r = −0.91 (p < 0.05), with the AMOC reduction lagging behind the sea ice extent increase by 4 years. The AMOC change lags behind Northern Hemisphere temperature change by 3 years (r = 0.92) and behind NGRIP temperature by 5 years (r = 0.70 ; both with p < 0.05). Northern Hemisphere temperatures lag behind NGRIP temperatures by 3 years (r = 0.84, p < 0.05). See **Figure R5** below.*

*However, the lagged correlations may not be relevant here as the time series are still noisy even with the running mean of 21 years. Lags are all zero when applying a larger smoothing of 51 years (before normalizing), or when restricting the calculation to the first 200 years after the perturbation is triggered.*

[Figure]

[Figure]

Figure R5 : (a) Normalized time series of the smoothed (blue) amoc, (orange) sea-ice extent, (green) Northern Hemisphere and (red) NGRIP temperatures with 21 years running mean (centered). (b) Correlations between time series for different lags [yr] (see figure legend). Lags are positive and leads are negative.

We will also replace lines 192 to 194 by the following :

*"The maximum changes for the sea-ice extent (+$10^{12}$ km² in comparison with the control), the AMOC (−5 Sv), the NGRIP (−0.53 °C) and Northern Hemisphere (−0.49 °C) temperatures are reached after 14, 18, 21 and 33 years respectively for the experiment associated with a large temperature anomaly intensity at the shelf drafts (i.e. X=100 ; Figure 5)."*

Line 204-206: can this be tested? What would you need to do to verify?
For simplicity, here is our detailed answer to reviewer #1 on this point.

More freshwater is added locally in the perturbed experiment than in the ctrl run at the end of the simulations (**Figure R6**), so our suggestion was wrong.

[Figure]

Figure R6 : Freshwater input [Sv] to the ocean model and annual sea-ice contour corresponding to 5,20,40,60 and 80% averaged over 470-500 years for (a) the control experiment and (b) the perturbation experiment with factor X=100.

We see that along the perturbation period, the yearly sea ice cover remains 0 after ~100 years. We also see that the temperature and salinity differences with the control occur over the whole water column at the chosen location (not only near the surface ; **Figure R7**).

[Figure]

Figure R7 : Hovmoller diagrams for the profile whose location is the blue dot in Figure R6b.
Absolute salinity [g/kg] and conservative temperature [°C] (a,b) for the perturbation experiment with factor X=100 and (c,d) the control simulation.

This led us to realize that the positive salinity and temperature anomalies, described lines 204-206, actually result from a resumption of the AMOC and renewed influx of heat and salt of Atlantic origin following the period of maximum AMOC slowdown. As we can see on the **Figure R8** below (that is the same as Figure 7 of the main text but with salinity anomalies included), the area of positive salinity anomaly at the end of the experiment is located not only at the surface but also at the subsurface along the Atlantic Water pathway.

[Figure]

Figure R8 : Conservative temperature [°C] at 300 m depth horizon, (a) averaged over the control experiment, and associated anomaly between the perturbation experiment with X=100 (b) over 10-40 yrs and (c) over 470-500 yrs and the control. (d,e,f) Same for the absolute salinity [g/kg].

This will be corrected in the revised manuscript, thank you for your comment. The revised sentence reads :

"*The positive anomalies therefore suggest a resumption of the AMOC and renewed influx of heat and salt of Atlantic origin following the period of maximum AMOC slowdown.*"

**Discussion:**

- Section 4.1: is your model too stable? Or is your model correct? How do you make this assessment? How might inaccuracies in your simulation affect this result? Is It limitations with dynamical ice sheet process simulation that make the model too stable, biases in simulated climate (atmosphere or ocean), uncertainty in other boundary conditions/forcings, differences in palaeogeographies, issues of resolution, or are the palaeo data simply misinterpreted or overly generalised? Thank you for these comments. We have evidence suggesting that the Hudson ice stream might have been very dynamic during the LGP, producing episodic large iceberg releases. Since our model does not reproduce this, even under large oceanic perturbations, we can assume that our model is too stable. This can come from ice sheet dynamical parameters and spatial resolution and/or climate biases (atmosphere and ocean). In particular, iLOVECLIM presents a warm atmospheric bias in North America that produces homogeneous temperate basal conditions (weak horizontal stresses). There is also a cold bias in the ocean in the Baffin Bay (**Figure R9**) that induces low thermal forcing in this area.

We will rephrase the beginning of this Section 4.1 as follows :

"*On the one hand, the ice sheets in the fully coupled model may be too stable. This could stem from the basal drag formulation, parameter values and spatial resolution. But it could also come from biases in the climate model, in the atmosphere (warm bias producing homogeneous temperate basal conditions) or in the ocean (low thermal forcing in the Baffin Bay).*

*On the other hand, we have pointed out that background temperatures are rather cold at the shelf drafts and salinity is low in the Baffin Bay and adjacent Labrador Sea in comparison with the same latitudes in the eastern part of the North Atlantic. Background oceanic basal melt rates are thus lower (around 0.003 m/yr) at the mouth of the Hudson ice stream in the control simulation than around the Fennoscandian ice sheet (around 0.3 m/yr). Therefore, in our simulations, the oceanic perturbation imposed by subsurface temperature amplification at the shelf drafts is not able to destabilize the North American ice sheet/streams even with the highest multiplicative factor.*

*Inaccurate representations of the ocean circulation or hydrography may contribute to underestimated ocean temperatures in the Baffin Bay area at 40 kyr BP. Such inaccuracies might come from the model low resolution (resulting in the misrepresentation of recirculation patterns for example) or from underestimated processes, such as the sinking of brines in the Southern Ocean, which could alter the meridional overturning circulation in the Atlantic (Bouttes et al., 2010). Improving the ocean representation is a key objective for future works. Indeed, our model can simulate carbon isotopes (Bouttes et al., 2015), allowing for direct comparison with observational data from marine sediment cores. This could serve as a constraint to improve ocean representation in subsequent studies."*

[Figure]

Figure R9 : Surface (a) conservative temperature from the World Ocean Atlas 2009 (1° x 1° spatial resolution), and for (b) iLOVECLIM Pre-industrial set-up (3° x 3° spatial resolution). Yellow, cyan, magenta and red contours are respectively 0, 1, 2 and 3 °C isocontours.

- Line 302: 'sensibly', how and why (i.e. caused by what)?
Since all other parameters remain constant, the differences between the two ice sheets arise from variations in bathymetry. Different bathymetries produce different ocean currents and advection patterns of heat and salt, resulting in different distributions of water masses (**Figure R10**). Furthermore, with the PI bathymetry, salinity is reduced because the salt content, that is the same as in the LGM, is more diluted due to the larger number of oceanic grid points. For instance, Barents and Kara Seas are present in the PI bathymetry (dark purple on **Figure R10**).

The primary differences between the two simulated ice sheets are observed around the Nordic Seas. With the PI bathymetry, the Fennoscandian ice sheet is smaller, while the Svalbard region shows a larger ice extent compared to the LGM bathymetry. Additionally, the southern extent of the Greenland ice sheet is reduced

and the North American ice sheet reaches lower latitudes on its eastern side under the PI bathymetry.

[Figure]

Figure R10: Maps of (a) conservative temperature [°C] and (b) absolute salinity [g/kg] near the surface, (c) bathymetry [m] together with ice sheet elevation [m] (shades of blue) for the control experiment with LGM bathymetry. (d,e,f) Same with the PI bathymetry.

Thank you for your comment. We will add the following sentence to comment on that :

*"The LGM and PI bathymetries produce different ocean currents and advection patterns of heat and salt, resulting in different distributions of water masses. Therefore, changing the bathymetry also produces a change in the ice sheet's geometry, leading here to a reduced volume of about 22 million km³ (44 mSLE). Differences in continental ice distribution include a North American ice sheet that reaches lower latitudes on its eastern side and a Fennoscandian ice sheet of smaller extent. Additionally, the PI bathymetry also leads to a reduction of the southern-east extent of the Greenland ice sheet."*

- Line 310: doesn't quite follow. What results in the greater anomaly? Simply that the model started with a smaller ice sheet? But that doesn't mean it will lose the same amount, it might lose less because less ice is in a vulnerable state with respect to your forcing. Line 311 doesn't make sense (in terms of English) – how can you compare LGM bathymetry with AMOC change? This all needs expanding, explicitly, the whole paragraph; too much is brushed over.

Thank you for your comment. We realize that this paragraph was not very clear. In this subsection, we have tried to refine the analysis using a different bathymetry. We did the same diagnosis but with a Pre-industrial bathymetry, as the 40 kyr BP bathymetry lies somewhere between LGM and PI bathymetries.

As explained above, different bathymetries produce different distributions of water masses as well as different ice sheet's geometries (**Figure R10**). Both these changes could lead to a different response of the coupled model to the oceanic perturbation experiment.

Due to the differing geometries and ice exposure to warmer waters in the Nordic Seas and in the Labrador Sea, the regions experiencing volume losses are not identical when we amplify the basal melt rates. This produces a difference in the amount of freshwater input to the ocean following the onset of the perturbation, though the difference with respect to the previous bathymetry is relatively small. The key climate variables (sea-ice extent, AMOC, Northern Hemisphere and NGRIP temperatures) follow similar trajectories to those obtained with the LGM bathymetry (Figure 5 of the main text and **Figure R11**).

Although we could expect a change in the fully coupled model response to the oceanic perturbation, following the change in bathymetry and distribution of the water masses, this is not the case here. The evolution of the feedback factor is also similar to what we obtained with the LGM bathymetry.

[Figure]

Figure R11: Same as Figure 5 in the main text but with the addition of the PI configuration.

We will rephrase the three final paragraphs (from line 305 to 320) as follows :

*"Due to the differing geometries and ice exposure to warmer waters in the Nordic Seas and in the Labrador Sea, the regions experiencing volume losses are not identical when we amplify the basal melt rates. The Greenland ice sheet loses significant volume along its southeastern coast, particularly at the beginning of the experiment with a large perturbation factor (X=100). The Eurasian ice sheet still experiences substantial losses in its northern regions. The Laurentide ice sheet undergoes dynamical thinning with this geometry, but these losses are mostly confined to its southeastern region, where it is exposed to warmer waters. This produces a relatively small difference in the amount of freshwater input to the ocean following the onset of the perturbation, with respect to the experiment performed with the LGM bathymetry. The key climate variables (sea-ice extent, AMOC, Northern Hemisphere and NGRIP temperatures) follow similar trajectories to those obtained with the LGM bathymetry.*

*Although we could expect a change in the fully coupled model response to the oceanic perturbation following the change in bathymetry and distribution of the water masses, this is not the case here. The evolution of the feedback factor is also similar to what we obtained with the LGM bathymetry."*

- Section 4.3: what about atmospheric dynamics? What are the model limitations? What is shown in the existing literature? How might your model representation affect your results? Is any of this relevant (and if not, why not)? We also come back to the comment above on model performance (point c on experiment design) – is that a relevant feature for this discussion section?
We thank you for this remark. We will add the following paragraphs at the end of Section 4.3 :

*"The atmospheric circulation was not a primary focus of this study, yet biases in the atmosphere model could also influence the ice sheet's volume variability and could be invoked to explain the absence of a DO-like event following the release of the perturbation. Climate biases have been assessed for the pre-industrial climate by Quiquet et al., (2021) : there is an overall cold bias around the Nordic Seas sector, Greenland and Fennoscandian ice sheets, with underestimated precipitation. In contrast, there is a relatively warm bias over North America, with underestimated precipitation except in mountainous areas where it tends to be overestimated (Quiquet et al., 2021). While it is unclear how these biases are translated at 40 kyr BP, they could affect the ice sheet's ability for regrowth and modulate the rate of its volume decay.*

*In addition, previous studies using the iLOVECLIM model have shown its ability to simulate abrupt temperature increases at the NGRIP site (~ +5 °C) associated with a significant rise in AMOC strength (~ +15 Sv) (Quiquet et al., 2021b, Figures 3 and 6). This suggests that, in our study, it is rather a lack of AMOC variability that prevents sudden warming after the perturbation is halted."*

 **Conclusion:**

I very much like this section, tight and punchy. You may want to revise/add to it following revisions in response to some of the comments above.
Thank you, **we will only modified point (3) and will add a new perspective to the conclusion, following the review process :**

**"*(3) The Laurentide ice sheet volume does not vary and the Hudson Strait ice stream is stable in our experiments with amplified basal melt rates. This can be partly explained by lower background temperature and salinity at shelf drafts in the Baffin Bay and Labrador Sea in comparison with the Nordic Seas. Imposing constant temperature offsets, up to 10 °C, produces a grounding line retreat and a volume decrease of the North American ice sheet, although the Hudson Strait region shows no dynamical instability. This suggests that the model is too stable in this region.***

*Different ice sheet geometries or modelling choices regarding the basal dynamics beneath the ice sheet could help to address the issue raised by point (3). This could be investigated in future work.*"

**Grammatical points/typos:**

Line 93: 'Were' -> 'Where'
Line 106: 'present' -> 'presents'
Line 124 'experiment' -> 'experiments'
Line 144: 'Eurasian ice sheet …' or 'Eurasia and Greenland…regain ice mass'.
Line 144: 'North American ice volume'
Line 153: 'Upstream from the'
Line 163: 'thickness decrease' -> 'thinning'?
Figure 6: update units on panels to match caption

Line 202: 'subsists' -> not sure this is the correct term to use here.
Line 213: remove first three 'the's (leave the fourth)
Line 229: 'allows us to highlight'
Line 252: 'there is less loss' or 'there are fewer losses'
Line 273: 'pointed out'
Line 292: needs rewording to improve the English
Line 298: 'terminating'
Line 314: 'at the beginning'
Thank you for pointing out these typos, they will be corrected.

Figure 10: Is there a problem with the rendering of the LGM line? I can't see it before ~250 yrs in the plot.
No, as indicated in the figure caption, the values are plotted only when the ice volume variation is above a critical value of $0.25 \cdot 10^{14}$ m³ to avoid 'division by 0' in Equation (3) as the grounded ice volume variations are not significant in the beginning of the simulations.

Thank you for all the comments and questions,

Louise Abot, on behalf of all co-authors.

**References**

Alvarez-Solas, Jorge, et al. "Links between ocean temperature and iceberg discharge during Heinrich events." *Nature Geoscience* 3.2 (2010): 122-126.

Álvarez-Solas, Jorge, et al. "Heinrich event 1: an example of dynamical ice-sheet reaction to oceanic changes." *Climate of the Past* 7.4 (2011): 1297-1306.

Alvarez-Solas, Jorge, et al. "Iceberg discharges of the last glacial period driven by oceanic circulation changes." *Proceedings of the National Academy of Sciences* 110.41 (2013): 16350-16354.

Arz, Helge W., et al. "Dominant Northern Hemisphere climate control over millennial-scale glacial sea-level variability." *Quaternary Science Reviews* 26.3-4 (2007): 312-321.

Barker, Stephen, et al. "Icebergs not the trigger for North Atlantic cold events." *Nature* 520.7547 (2015): 333-336.

Bond, Gerard, et al. "Evidence for massive discharges of icebergs into the North Atlantic ocean during the last glacial period." *Nature* 360.6401 (1992): 245-249.

Bond, Gerard, et al. "Correlations between climate records from North Atlantic sediments and Greenland ice." *Nature* 365.6442 (1993): 143-147.

Bond, Gerard C., and Rusty Lotti. "Iceberg discharges into the North Atlantic on millennial time scales during the last glaciation." *science* 267.5200 (1995): 1005-1010.

Batchelor, Christine L., et al. "The configuration of Northern Hemisphere ice sheets through the Quaternary." *Nature communications* 10.1 (2019): 3713.

Berger, André. "Long-term variations of caloric insolation resulting from the Earth's orbital elements." *Quaternary research* 9.2 (1978): 139-167.

Bouttes, Nathaëlle, D. Paillard, and Didier M. Roche. "Impact of brine-induced stratification on the glacial carbon cycle." *Climate of the Past* 6.5 (2010): 575-589.

Bouttes, Nathaëlle, et al. "Including an ocean carbon cycle model into iLOVECLIM (v1. 0)." *Geoscientific Model Development* 8.5 (2015): 1563-1576.

Crotti, Ilaria, et al. "Wilkes subglacial basin ice sheet response to Southern Ocean warming during late Pleistocene interglacials." *Nature Communications* 13.1 (2022): 5328.

Dalton, April S., Chris R. Stokes, and Christine L. Batchelor. "Evolution of the Laurentide and Innuitian ice sheets prior to the Last Glacial Maximum (115 ka to 25 ka)." *Earth-Science Reviews* 224 (2022): 103875.

Dansgaard, Willi, et al. "North Atlantic climatic oscillations revealed by deep Greenland ice cores." *Climate processes and climate sensitivity* 29 (1984): 288-298.

Dansgaard, Willi, et al. "Evidence for general instability of past climate from a 250-kyr ice-core record." *nature* 364.6434 (1993): 218-220.

Ditlevsen, Peter D., Katrine Krogh Andersen, and Anders Svensson. "The DO-climate events are probably noise induced: statistical investigation of the claimed 1470 years cycle." *Climate of the Past* 3.1 (2007): 129-134.

Elliot, Mary, et al. "Millennial‐scale iceberg discharges in the Irminger Basin during the last glacial period: Relationship with the Heinrich events and environmental settings." *Paleoceanography* 13.5 (1998): 433-446.

Friedrich, T., et al. "The meganism behind internally generated centennial-to-millennial scale climate variability in an earth system model of intermediate complexity." *Geoscientific Model Development* 3.2 (2010): 377-389.

Gottschalk, Julia, et al. "Abrupt changes in the southern extent of North Atlantic Deep Water during Dansgaard–Oeschger events." *Nature geoscience* 8.12 (2015): 950-954.

Gowan, Evan J., et al. "A new global ice sheet reconstruction for the past 80 000 years." *Nature communications* 12.1 (2021): 1199.

Gwyther, David E., et al. "Modelling the response of ice shelf basal melting to different ocean cavity environmental regimes." *Annals of Glaciology* 57.73 (2016): 131-141.

Heinrich, Hartmut. "Origin and consequences of cyclic ice rafting in the northeast Atlantic Ocean during the past 130,000 years." *Quaternary research* 29.2 (1988): 142-152.

Henry, L. G., et al. "North Atlantic ocean circulation and abrupt climate change during the last glaciation." *Science* 353.6298 (2016): 470-474.

Johnsen, S. J., et al. "Irregular glacial interstadials recorded in a new Greenland ice core." *Nature* 359.6393 (1992): 311-313.

Jonkers, Lukas, et al. "A reconstruction of sea surface warming in the northern North Atlantic during MIS 3 ice-rafting events." *Quaternary Science Reviews* 29.15-16 (2010): 1791-1800.

Kindler, Philippe, et al. "Temperature reconstruction from 10 to 120 kyr b2k from the NGRIP ice core." *Climate of the Past* 10.2 (2014): 887-902.

Lüthi, Dieter, et al. "High-resolution carbon dioxide concentration record 650,000–800,000 years before present." *nature* 453.7193 (2008): 379-382.

Kleman, Johan, et al. "North American Ice Sheet build-up during the last glacial cycle, 115–21 kyr." *Quaternary Science Reviews* 29.17-18 (2010): 2036-2051.

Marshall, Shawn J., et al. "Glaciological reconstruction of the Laurentide Ice Sheet: physical processes and modelling challenges." *Canadian Journal of Earth Sciences* 37.5 (2000): 769-793.

Max, Lars, et al. "Subsurface ocean warming preceded Heinrich Events." *Nature Communications* 13.1 (2022): 4217.

Rasmussen, Tine L., and Erik Thomsen. "The role of the North Atlantic Drift in the millennial timescale glacial climate fluctuations." *Palaeogeography, Palaeoclimatology, Palaeoecology* 210.1 (2004): 101-116.

Schulz, Michael. "On the 1470‐year pacing of Dansgaard‐Oeschger warm events." *Paleoceanography* 17.2 (2002): 4-1.

MacAyeal, D. R. "Binge/purge oscillations of the Laurentide ice sheet as a cause of the North Atlantic's Heinrich events." *Paleoceanography* 8.6 (1993): 775-784.

Makinson, Keith, et al. "Influence of tides on melting and freezing beneath Filchner‐Ronne Ice Shelf, Antarctica." *Geophysical Research Letters* 38.6 (2011).

Margold, Martin, Chris R. Stokes, and Chris D. Clark. "Ice streams in the Laurentide Ice Sheet: Identification, characteristics and comparison to modern ice sheets." *Earth-Science Reviews* 143 (2015): 117-146.

Mathiot, Pierre, et al. "Explicit representation and parametrised impacts of under ice shelf seas in the z∗ coordinate ocean model NEMO 3.6." *Geoscientific Model Development* 10.7 (2017): 2849-2874.

Mignot, Juliette, Andrey Ganopolski, and Anders Levermann. "Atlantic subsurface temperatures: Response to a shutdown of the overturning circulation and consequences for its recovery." *Journal of Climate* 20.19 (2007): 4884-4898.

Miller, Gifford H., and John T. Andrews. "Hudson Bay was not deglaciated during MIS-3." *Quaternary Science Reviews* 225 (2019): 105944.

Pico, T., et al. "Refining the Laurentide Ice Sheet at Marine Isotope Stage 3: A data-based approach combining glacial isostatic simulations with a dynamic ice model." *Quaternary Science Reviews* 195 (2018): 171-179.

Quiquet, Aurélien, et al. "The GRISLI ice sheet model (version 2.0): calibration and validation for multi-millennial changes of the Antarctic ice sheet." *Geoscientific Model Development* 11.12 (2018): 5003-5025.

Quiquet, Aurélien, et al. "Climate and ice sheet evolutions from the last glacial maximum to the pre-industrial period with an ice-sheet–climate coupled model." *Climate of the Past* 17.5 (2021): 2179-2199.

Schoof, Christian. "Ice sheet grounding line dynamics: Steady states, stability, and hysteresis." *Journal of Geophysical Research: Earth Surface* 112.F3 (2007).

Siddall, Mark, et al. "Marine isotope stage 3 sea level fluctuations: Data synthesis and new outlook." *Reviews of Geophysics* 46.4 (2008).

Waelbroeck, Claire, et al. "Sea-level and deep water temperature changes derived from benthic foraminifera isotopic records." *Quaternary science reviews* 21.1-3 (2002): 295-305.

Waelbroeck, Claire, et al. "Relative timing of precipitation and ocean circulation changes in the western equatorial Atlantic over the last 45 kyr." *Climate of the Past* 14.9 (2018): 1315-1330.

---

## Author Response (AR3)

**Response to the Editor**

The reviewers appreciate the effort put into addressing their comments and now judge it acceptable for publication (once the minor comments from Referee # 1 are addressed). In my read through, I have noted a number of easy to address issues. Unless you disagree with any of the comments, the response to editor need only be "all comments addressed" along with a latex-diff (or tracked changes) showing the requested changes/clarifications/corrections.

Thank you for the time you spent reading the manuscript and your comments, all of which have been addressed.

Regarding your comment on conversion from m3 to mSLE, we would like to clarify that we do not use a conversion factor to derive the ice volume in mean sea level equivalent. Instead, we isolate the grounded volume that is above sea level (from the total volume, in millions of km3 in the main text) that we then divide by the present-day ocean area, after multiplication by the density ratio. We have added a sentence in the main text to clarify this point.

Sincerely yours,

Louise Abot, on behalf of all co-authors

**Response to Reviewer #1's comment**

I would like to thank the authors for their thorough response to my comments. They have clarified some important issues and substantiated their results further with many additional model simulations.

I only have a few very minor additional comments on the revised manuscript (line numbers refer to the tracked changes version):

L. 40 and 43: The definition of the Last glacial period as between does not seem standard and needs a reference. Why would the glacial period terminate at the LGM? Thank you for your comment, we have added references and corrected an error concerning the end date of the LGM.

L. 40-43: I think that here it would be important to mention already that millennial-scale climate variability (DO events) associated with transitions in the AMOC is expected to affect sub-surface temperatures. Maybe just move up lines 59-62? Thank you, we have moved up these lines.

Fig. 1b,c: the contour lines make the panels a bit difficult to read. Maybe use a style similar to Fig. 3e or f instead? Thank you, we have reworked this figure with lighter contours.

Fig. 4: why are the calving and basal melt fluxes reported as negative values? It was an arbitrary convention to count negatively what goes out of the cap. We have changed it. Thank you for this comment.

L. 197: HS5 -> HS4 Thank you, we corrected it.

L. 201: the 220 ppm should be either the CO2 concentration or an equivalent radiative CO2 concentration accounting also for the radiative forcing of other GHGs Thank you, it was a mistake, it has been corrected.

L. 368: lower -> weaker? Thank you, we changed it.

Thank you again for your careful reading and all the comments.

Sincerely yours,

Louise Abot, on behalf of all co-authors